# Distinct roles for H4 and H2A.Z acetylation in RNA transcription in African trypanosomes

Amelie J. Kraus [ID] [1,2,3,5], Jens T. Vanselow[4,6], Stephanie Lamer[4], Benedikt G. Brink [ID] [1,2], Andreas Schlosser[4] & T. Nicolai Siegel [ID] [1,2,3 ✉]

Despite histone H2A variants and acetylation of histones occurring in almost every eukaryotic organism, it has been difficult to establish direct functional links between canonical histones or H2A variant acetylation, deposition of H2A variants and transcription. To disentangle these complex interdependent processes, we devised a highly sensitive strategy for quantifying histone acetylation levels at specific genomic loci. Taking advantage of the unusual genome organization in *Trypanosoma brucei*, we identified 58 histone modifications enriched at transcription start sites (TSSs). Furthermore, we found TSS-associated H4 and H2A.Z acetylation to be mediated by two different histone acetyltransferases, HAT2 and HAT1, respectively. Whereas depletion of HAT2 decreases H2A.Z deposition and shifts the site of transcription initiation, depletion of HAT1 does not affect H2A.Z deposition but reduces total mRNA levels by 50%. Thus, specifically reducing H4 or H2A.Z acetylation levels enabled us to reveal distinct roles for these modifications in H2A.Z deposition and RNA transcription.

[1] Department of Veterinary Sciences, Experimental Parasitology, Ludwig-Maximilians-Universität München, 80752 Munich, Germany. [2] Biomedical Center Munich, Department of Physiological Chemistry, Ludwig-Maximilians-Universität München, 82152 Planegg-Martinsried, Germany. [3] Research Center for Infectious Diseases, University of Würzburg, 97080 Würzburg, Germany. [4] Rudolf Virchow Center for Experimental Biomedicine, University of Würzburg, 97080 Würzburg, Germany. [5] Present address: Institute for Epigenetics and Stem Cells, Helmholtz Zentrum München, Munich, Germany. [6] Present address: German Federal Institute for Risk Assessment, Unit Safety of Chemicals, Department Chemicals and Product Safety, Berlin, Germany. ✉email: n.siegel@lmu.de

Genomic DNA is the ultimate template for our heredity and therefore must be faithfully replicated, repaired, and its information transcribed into RNA. These processes require that large multi-protein complexes are able to access the DNA. However, most eukaryotic DNA is packaged into compact chromatin structures composed of DNA and proteins. While this organization can form an obstacle to DNA-templated processes, it also provides an opportunity for regulation because dynamic chromatin structures can be locally and globally modified to influence DNA accessibility[1].

Structural changes in chromatin can be induced by post-translational modifications (PTMs) of histones or by the replacement of canonical histones with histone variants. Among the best-characterized histone PTMs are acetylation, phosphorylation, and methylation. These PTMs can alter the conformation of nucleosomes or function as specific binding sites for enzymatic complexes, which subsequently alter the chromatin structure (reviewed in ref. [2]). For example, acetylation of the N-terminal tail of histone H4 weakens inter-nucleosomal contacts, which disrupt the higher-order folding of chromatin fibers, thereby contributing to a more open chromatin conformation that favors active transcription[3]. Alternatively, histone modifications can alter the chromatin structure by providing binding sites for histone-binding proteins. These proteins contain characteristic domains, such as chromo- or bromodomains, that are involved in binding to distinct histone modifications, e.g., methylated or acetylated lysine residues, respectively[4,5]. In agreement with the observed correlation between hyperacetylation and actively transcribed genes, many transcription factors, including histone acetyltransferases (PCAF, GCN5, and TAFII250) and chromatin remodeling complexes (SWR1, SWI/SNF, RSC) contain bromodomains[6,7]. Therefore, once a site is acetylated and poised for transcription, it can promote further recruitment of histone acetyltransferases (HATs) leading to additional acetylation. Thus, local effects of histone acetylation that generate a more open chromatin conformation can be further enhanced by the recruitment of chromatin remodeling complexes via acetylation sites (reviewed in ref. [8]).

Chromatin structure can also be altered by replacing canonical histones with histone variants that differ in primary amino acid sequence from their canonical paralogues. For each of the canonical histones exists at least one variant. While some of these variants are responsible for very specialized functions in certain species, others, such as H3.3, CENP-A or H2A.Z, are widely conserved in evolution. Thus far, absence of H2A.Z has only been reported for Giardia, Trichomonas and Entamoeba spp., making it the most widely conserved histone variant and suggesting an important and potentially conserved function[9,10]. H2A.Z is synthesized and deposited throughout the cell cycle and can influence a multitude of biological processes including transcription, DNA repair and replication, chromosome segregation, and suppression of antisense RNA[11]. Like other histones, H2A.Z can be post-translationally modified and changes in PTM patterns might allow H2A.Z to perform such a wide variety of functions and to have potentially opposite roles[12]. For example, while H2A.Z has been found at inactive and active promoters[13], acetylated H2A.Z (H2A.Zac) appears to be enriched only at promoters of active genes and absent from inactive genes[14,15]. Thus, H2A.Zac is probably more important for transcriptional activation than unmodified H2A.Z. However, cause and consequence of H2A.Zac enrichment at promoters has been difficult to study in vivo, since in yeast and mammals the HAT responsible for H2A.Z acetylation is also the main enzyme responsible for H4 tail acetylation, a prerequisite for H2A.Z deposition[16].

Global chromatin profiling approaches such as ChIP-seq have provided a detailed picture of the genome-wide distribution of many PTMs and histone variants. From numerous such analyses it is evident that specific PTMs or histone variants are enriched at distinct genomic features. For example, in human CD4+ T cells H3K4 methylation is enriched at transcription start sites (TSSs) of protein-coding genes, while H2A.Z is found upstream and downstream of TSSs[17]. It is thought that at TSSs, acetylated H2A.Z, in combination with other factors, leads to increased DNA accessibility, thereby facilitating the access of the transcription complex to specific DNA sequence elements[16].

Despite this growing knowledge of the genome-wide distribution of PTMs and histone variants, we know relatively little about the mechanisms involved in their targeted deposition at specific genomic loci[18].

Deposition of histone variants is performed by specific ATP-dependent chromatin remodeling complexes or by specialized histone chaperones in a specific or un-specific manner[7]. For example, in yeast targeted deposition of H2A.Z (Htz1) has been observed at sites containing acetylated histones or a specific DNA sequence motif[13,19]. In addition, it has been proposed that un-specifically deposited H2A.Z is subsequently cleared by the transcription machinery or chromatin remodellers[20]. While the specific and un-specific deposition of H2A.Z do not have to occur in a mutually exclusive manner, the observation that both mechanisms can exist highlights the complexity of chromatin-mediated gene expression. In budding yeast, the 14 subunits-containing SWR1 complex is responsible for most H2A.Z deposition and homologous complexes have been identified in more complex eukaryotes. Interestingly, all of these complexes contain a bromodomain factor, for example Bdf1 in yeast and Bdf8 in humans (reviewed in ref. [7]), suggesting an intrinsic link between H2A.Z deposition and histone acetylation. Indeed, using yeast it was shown in vitro that acetylation of histone H4 tails stimulates deposition of H2A.Z by the SWR1 complex[21]. Most eukaryotic genomes contain thousands of TSSs of which some are transcriptionally active at any given time while others are not. This complexity has made it difficult to study the links between H4 acetylation and H2A.Z deposition in vivo on a genome-wide scale and to determine how the absence of H4 or H2A.Z acetylation affects transcription initiation.

As in other eukaryotes, DNA in the protozoan parasite Trypanosoma brucei is packaged into chromatin. Even though the primary sequences of trypanosome core histones diverge significantly from those in other eukaryotes, several histone modifications have been reported[22–25], including an extensively acetylated H4 tail. Furthermore, trypanosomes contain one variant form of each of the four core histones[23,26] with TSSs being enriched with H2A.Z, acetylated histones and different BDFs[27,28].

While ChIP-seq studies specifically enrich for nucleosomes containing specific PTMs or specific histone variants, understanding how a specific histone variant is targeted to a specific genomic locus requires the reverse analysis, i.e., being able to determine all PTMs present on the chromatin isolated from a specific genomic locus. Such locus-specific chromatin isolations followed by mass spectrometry-based analyses can identify all factors involved in the establishment of a locus-specific chromatin structure, PTMs, histone variants, histone-modifying enzymes, and chromatin remodeling complexes. However, the success of such locus-specific chromatin isolations greatly depends on the ability to enrich chromatin from specific loci over chromatin from other regions. Thus far, only chromatin from long repetitive regions such as telomeric repeats has been successfully studied by mass-spectrometry[29].

To overcome this problem, we have taken advantage of the unusual genome organization in trypanosomes. Atypically for a eukaryote, most RNA Pol II transcribed genes in trypanosomes are arranged in polycistronic transcription units (PTUs)[30] that

are preceded by large (~10 kb) TSSs, containing ~50 H2A.Z-nucleosomes[27]. Furthermore, ChIP-seq data suggest that all TSSs are marked by identical chromatin structures with similar levels of histone acetylation and H2A.Z present at each TSS[27]. Since *T. brucei* contain relatively little non-coding DNA, TSSs contribute to ~7% of the total *T. brucei* genome[31], making the parasite a valuable model to study TSS-associated chromatin.

Thus, using *T. brucei*, the goal of this study was to understand how TSS-specific chromatin structures are established and how their absence affects transcription. Therefore, we established an approach for locus-specific chromatin analyses that allowed us to obtain a comprehensive picture of PTMs present in *T. brucei*.

Building on our previously reported approach to quantify histone acetylation levels[32], we determined histone acetyl marks enriched at TSSs and found TSS-associated H4 and H2A.Z acetylation to be mediated by the two MYST acetyltransferases HAT1 and HAT2. Depletion of HAT2 leads to a loss of TSS-associated H4 acetylation, a loss in H2A.Z deposition and a shift in RNA Pol II transcription initiation sites. In contrast, depletion of HAT1 only has a minor effect on H2A.Z deposition but leads to reduced H2A.Z acetylation and a global decrease in transcript levels. Thus, our study allowed us to disentangle the processes of H2A.Z deposition and acetylation revealing a direct link between histone acetylation, H2A.Z deposition and RNA Pol II transcription initiation in an evolutionarily highly divergent eukaryote.

## Results

**Isolation of nucleosomes from TSSs.** Previously, we found a single acetyl mark (H4K10ac, which is potentially functionally equivalent to H4K12ac in other eukaryotes) to be enriched at TSSs in *T. brucei*[27]. The presence of other TSS-specific modifications has not been evaluated due to a lack of specific antibodies and no functional link could be established between histone acetylation, H2A.Z deposition or RNA transcription.

To identify all PTMs enriched at TSSs in an unbiased manner, we sought to isolate nucleosomes from transcription start sites (TSS-nucleosomes) and nucleosomes not located at start sites (non-TSS-nucleosomes) (Supplementary Fig. 1a) and to compare their PTM patterns. Previously published ChIP-seq[27] and co-immunoprecipitation studies[33] point to a largely mutually exclusive distribution of H2A.Z- and H2A-containing nucleosomes in *T. brucei*, with H2A.Z being highly enriched at TSSs and not detectable in H2A-containing nucleosomes.

To validate these observations, we repeated the ChIP-seq analysis of H2A.Z and performed ChIP-seq to determine the distribution of TY1-tagged H2A. The strong depletion of H2A at TSSs suggests that the distribution of H2A.Z- and H2A-containing nucleosomes is indeed to a large extent mutually exclusive (Fig. 1a). Next, to

determine whether the observed distribution would allow us to specifically isolate nucleosomes containing only H2A.Z or H2A, we immunoprecipitated H2A.Z- and H2A-containing nucleosomes and analyzed the co-immunoprecipitated histones by western blotting. No H2A could be detected following immunoprecipitation of H2A.Z and, as reported before[33], no H2A.Z could be detected following the immunoprecipitation of H2A (Fig. 1b). These findings point to very low levels of heterotypic H2A.Z/H2A nucleosomes, which have been observed in other organisms[34], and should allow us to enrich for nucleosomes from TSSs (containing H2A.Z) and non-TSSs (containing H2A).

To enrich for TSS-nucleosomes, we used a previously generated cell line expressing only TY1-tagged H2A.Z. In two ways we confirmed that TY1-tagging and overexpression of H2A.Z does not affect H2A.Z localization. First, we performed ChIP-seq of TSS-nucleosomes using either a custom-made antibody against untagged H2A.Z or an antibody recognizing the TY1-tag and found that overexpression of a TY1-tagged H2A.Z did not affect its genome-wide distribution (Supplementary Fig. 1b). Next, we used mass-spectrometry to analyze H2B.V co-immunoprecipitated with TY1-tagged H2A.Z or with untagged H2A.Z and obtained identical PTM patterns from both IPs (Supplementary Fig. 1c). Since H2B.V has been shown to dimerize exclusively with H2A.Z in *T. brucei*[33], these observations suggest that the TY1-tag does not affect the H2A.Z distribution nor the PTM-pattern of TSS-nucleosomes. Thus, we isolated H2A.Z-containing nucleosomes by immunoprecipitation to identify PTMs enriched at TSSs (outlined in Supplementary Fig. 2).

As our ChIP-seq suggested H2A to be largely depleted from TSSs, we decided to isolate non-TSS-nucleosomes by immunoprecipitation of H2A. Since H2A genes exist in a large multicopy family, we used a previously published cell line expressing TY1-tagged H2A in addition to endogenous, untagged H2A[33]. Despite the presence of untagged H2A, TY1-tagged H2A efficiently incorporated into the genome, as suggested by the robust co-immunoprecipitation of the other canonical histones (Supplementary Fig. 3). By taking advantage of the distinct genome-wide distributions of H2A.Z and H2A, we were able to specifically enrich TSS-nucleosomes (those containing H2A.Z) and non-TSS-nucleosomes (those containing H2A).

**Nucleosomes at TSSs are hyperacetylated.** Having separately enriched TSS- and non-TSS-nucleosomes, we sought to analyze their specific PTM patterns. Given the link between histone acetylation and H2A.Z deposition[13,19,21,35–37], we initially focused on acetyl marks and applied Fragment Ion Patchwork Quantification (FIPQuant), a previously established method to accurately quantify the site-specific levels of histone acetylation

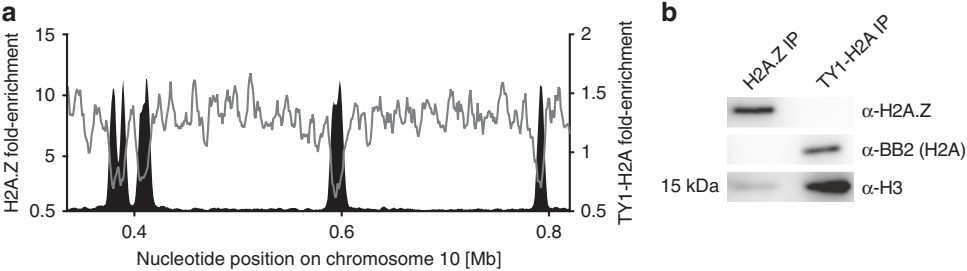

**Fig. 1 H2A.Z replaces H2A at TSSs. a** MNase-ChIP-seq data of the histone variant H2A.Z (black; $n = 1$) and of the overexpressed and TY1-tagged canonical histone H2A (gray; $n = 2$) are shown across a representative region of chromosome 10. The ChIP-seq data are normalized to input data. **b** Western blot of proteins co-immunoprecipitated with H2A.Z (left lane) or TY1-H2A (right lane). Mononucleosomes were extracted from a cell line, which allows the inducible overexpression of ectopic TY1-H2A, and were immunoprecipitated using either a custom-made antibody against untagged H2A.Z ($n = 1$) or the BB2 antibody recognizing TY1-tagged H2A ($n = 1$). Immunoprecipitated material was then examined by western blotting for the presence of TY1-H2A or untagged H2A.Z along with histone H3 as loading control. Source data are provided as a Source Data file.

(Fig. 2a)[32]. The ability to quantify acetyl levels should allow us to identify marks enriched at TSS-nucleosomes or non-TSS-nucleosomes even if H2A.Z and H2A distribution is not fully mutually exclusive.

The use of different unspecific proteases (elastase, thermolysin, papain) in FIPQuant yielded full coverage of histone sequences and allowed us to perform a comprehensive screen for different PTMs. For a semi-quantitative analysis of histone methyl marks, we relied on spectra counting (see Methods). Analyzing TSS-nucleosomes using FIPQuant, we confirmed the previously reported H4K10ac mark to be enriched at TSSs, validating our approach. In addition, our analysis revealed acetyl marks at A1 (N-terminal α-amino group), K2 and K5 to be strongly enriched at TSSs (Fig. 2b and Supplementary Fig. 4). As expected, based on its high abundance, H4K4ac was acetylated at both TSS- and non-TSS-nucleosomes (Fig. 2b). Whereas H3 is frequently acetylated in most other eukaryotes, we only found H3K23 to be acetylated in *T. brucei* and that only at TSSs (Fig. 2b).

In addition to H4, H2A.Z and H2B.V were also highly acetylated. H2A.Z contains a much longer N-terminal tail than H2A (41 aa compared to 1 aa; Supplementary Fig. 5) and we found the two lysines on the N-terminal tail and the seven adjacent lysines on the histone-fold domain to be hyperacetylated (Fig. 3 and Supplementary Fig. 6). Similarly, we found the N-terminal portion of H2B.V to contain a large number of highly acetylated lysines (Fig. 3 and Supplementary Fig. 6).

On histone H3, we also detected site-specific methylation patterns. H3 from non-TSS-nucleosomes only carried mono-, di- and trimethylation at H3S1 and H3K76. By contrast, H3 from TSS-nucleosomes also carried all three methylation states at positions H3K4, H3K10, and H3K11 (Fig. 2b and Supplementary Fig. 4). This is in good agreement with H3K4me3 ChIP-seq data that indicate an enrichment of H3K4me3 at TSSs[38]. In addition, H3 from non-TSS-nucleosomes carried mono-methyl marks at H3K4, H3K10, H3K16, and H3K19. For histone H4 methylation, we did not observe notable differences between TSSs and non-TSSs, except for H4K2 monomethylation, which was TSS-specific.

In summary, by performing FIPQuant of *T. brucei* histones we found that TSS-nucleosomes are more extensively acetylated compared to non-TSS-nucleosomes. Furthermore, we discovered several PTM patterns not previously observed in other eukaryotes.

**_T. brucei_ histones have more than 150 modifications**. Previous attempts to identify and map PTMs in *T. brucei* revealed the presence of several acetyl and methyl marks. However, for technical reasons these studies failed to obtain full coverage of core histone sequences[22,23]. In addition, no PTMs had been reported for histone variants in *T. brucei*. Our immunoprecipitation-based isolation of histones enabled the identification of a large number of PTMs, many of which had not been identified in *T. brucei* before. However, analysis of histones co-immunoprecipitated with H2A.Z or H2A might fail to detect PTMs only found on free histones or in very compacted chromatin.

Therefore, to obtain a comprehensive list of PTMs present in *T. brucei*, we complemented our analysis of immunoprecipitated histones with an analysis of histones isolated by acid extraction from whole cell lysates (outlined in Supplementary Fig. 2). Combining the results from the different histone isolation strategies and only counting PTMs that were identified in at least three separate experiments, we detected 40 acetylation, 54 mono-, 33 di-, 26 trimethylation and 4 phosphorylation marks (Fig. 4 and Supplementary Fig. 7). Six PTMs, mostly methyl marks, were only found on histones isolated by acid extraction

(Fig. 3 and Supplementary Fig. 6). While we detected a large number of acetyl marks on TSS-nucleosomes, almost no acetylation was observed on the histone variants H3.V or H4.V (Fig. 3b and Supplementary Fig. 6b). Both of these variants have been linked to transcription termination in *T. brucei* and are absent from TSS-nucleosomes[27,39,40].

Despite the large number of PTMs identified in this study, we did not detect some of the previously reported modifications (Fig. 4 and Supplementary Data 1). Examples include the low abundant (~1%) acetyl marks on the N-terminal tail of H2B (K4, K12 and K16) (Fig. 3a and Supplementary Fig. 6a), although we did find several acetyl marks on the N-terminal tail of H2B.V (Fig. 3b and Supplementary Fig. 6b).

Combined, our data represent the first comprehensive analysis of histone modifications in *T. brucei*, fully covering the sequences of all four histones and four histone variants (Fig. 3 and Supplementary Fig. 6). In total, we identified 157 PTMs (223, if we include PTMs also identified in less than three independent experiments), 126 of which have not been reported before in *T. brucei*, including prominent methyl marks on the N-terminal tail of H3 (Supplementary Data 1). While the total number of PTMs identified in the study is lower than that reported for *Saccharomyces cerevisiae* histones[41], the extent of histone modifications is much higher than what had been anticipated for *T. brucei* given its small number of putative histone-modifying enzymes[23,26] and its apparent lack of RNA Pol II transcription regulation[42].

**HAT1 and HAT2 are responsible for distinct acetylation marks**. In budding yeast, Esa1, the catalytic component of the NuA4 histone acetyltransferase, is the main enzyme responsible for acetylation of histone H4[43]. Similarly, in humans the chromatin remodeling complex p400 is responsible for the deposition of H2A.Z and contains an acetyltransferase (Tip60) that is implicated in the acetylation of histone H4[44]. However, both Esa1 and Tip60 are not only responsible for acetylation of H4 but also for the acetylation of H2A.Z, complicating the establishment of a functional link between histone acetylation and H2A.Z deposition in vivo[16]. To determine the contribution of H4 and H2A.Z acetylation to H2A.Z deposition in vivo, we sought to identify the enzymes responsible for the TSS-specific H2A.Z, H2B.V, and H4 (K2, K5, and K10) acetyl marks in *T. brucei* and to determine whether loss of these marks impacts H2A.Z deposition.

The *T. brucei* genome encodes six different HATs[26,45,46]. Like the NuA4 acetyltransferase, HAT1-3 are related to the MYST-family acetyltransferases and only HAT1 and HAT2 appear to be essential for parasite growth[46–48]. In addition, a systematic target site screen using FIPQuant for HAT3 revealed that this enzyme is responsible for acetylation of a single residue: H4K4[32], which is, according to our data, the only acetyl mark on H4 that is not enriched at TSSs. By contrast, a systematic analysis of HAT1 and HAT2 has not been performed meaning that their target sites are still unknown, with the exception of HAT2-mediated H4K10 acetylation[46].

To reveal the target sites of HAT1 and HAT2, we depleted HAT1 or HAT2 in *T. brucei* using RNAi, isolated histones from mononucleosomes by acid extraction or immunoprecipitated TSS-nucleosomes and performed FIPQuant to detect changes in the acetylome. An RNAi system which conditionally expresses intramolecular stem-loop, double-stranded RNA[49] was used to generate cell lines for inducible knockdown of HAT1 and HAT2 transcripts. Following 48 h of RNAi induction, HAT1 and HAT2 transcript levels were reduced to 9% and 33% of wild-type levels, respectively (Supplementary Fig. 8a and Supplementary Data 2). As reported previously, depletion of

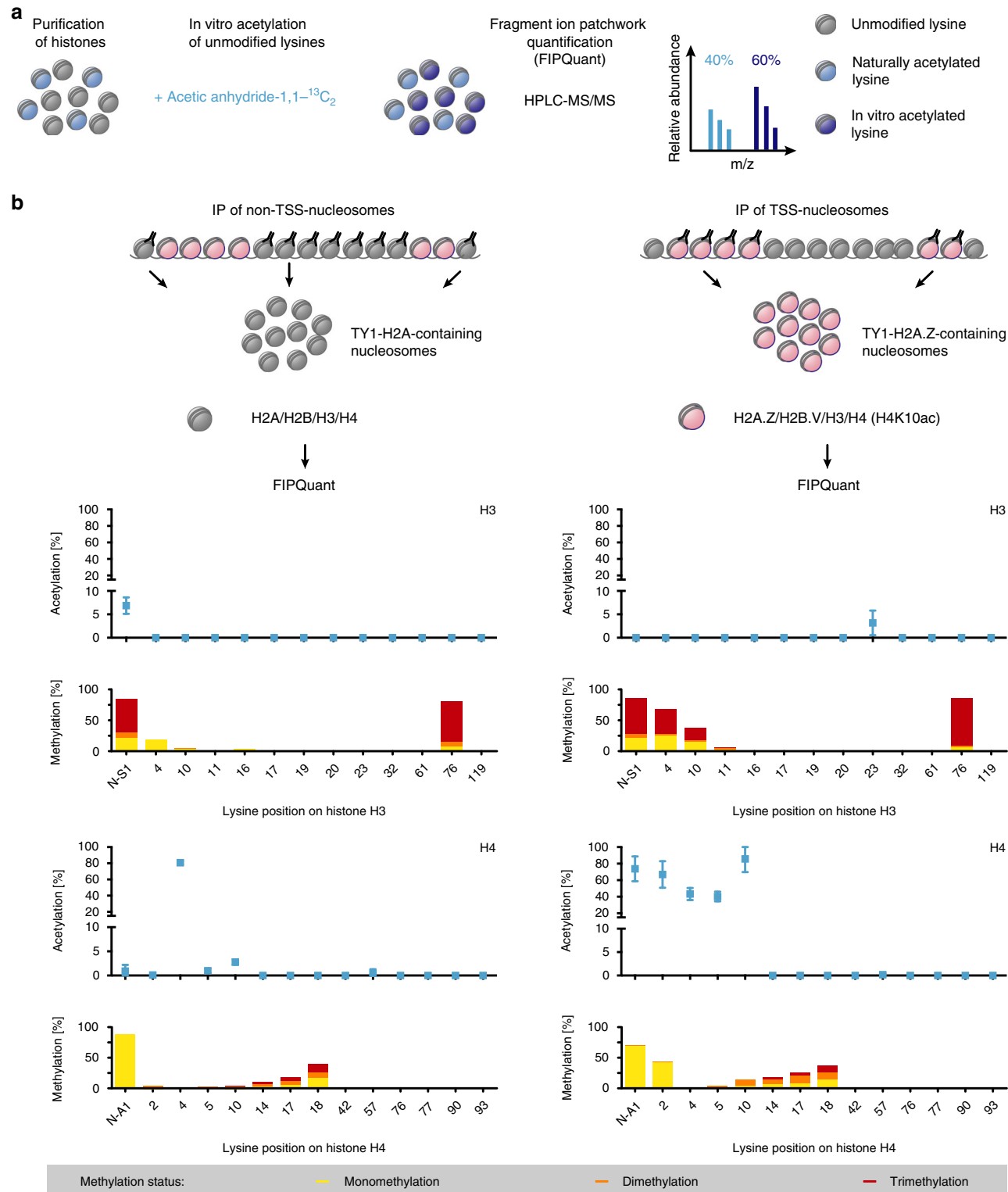

**Fig. 2 Identification of acetyl and methyl marks enriched at TSSs. a** Outline of the Fragment Ion Patchwork Quantification (FIPQuant) methodology. Following histone purification, unmodified lysines were in vitro labeled with a $C_{13}$-acetyl mark and analyzed by mass spectrometry. The levels of site-specific lysine acetylation were determined using FIPQuant. **b** PTM-patterns of H3 and H4 from non-TSS- and TSS-nucleosomes. TSS- and non-TSS-nucleosomes were enriched by immunoprecipitation of TY1-H2A.Z and TY1-H2A-containing nucleosomes, respectively. The acetylation percentages [%] (blue) represent the averages of the median values from each of the independent experiments (left panel $n = 3$, right panel $n = 7$) determined by FIPQuant. Error bars indicate standard deviations. Lysine-specific mono- (yellow), di- (orange) and/or trimethylation (red) levels are shown for H3 and H4 from non-TSS- (left panel; $n = 3$) and TSS-nucleosomes (right panel; $n = 7$) and are plotted as stacked bars representing the averages of the estimated methylation percentages based on identified mono-, di- and trimethylated spectra. Supplementary Fig. 4 shows the data of each replicate. Source data are provided as a Source Data file.

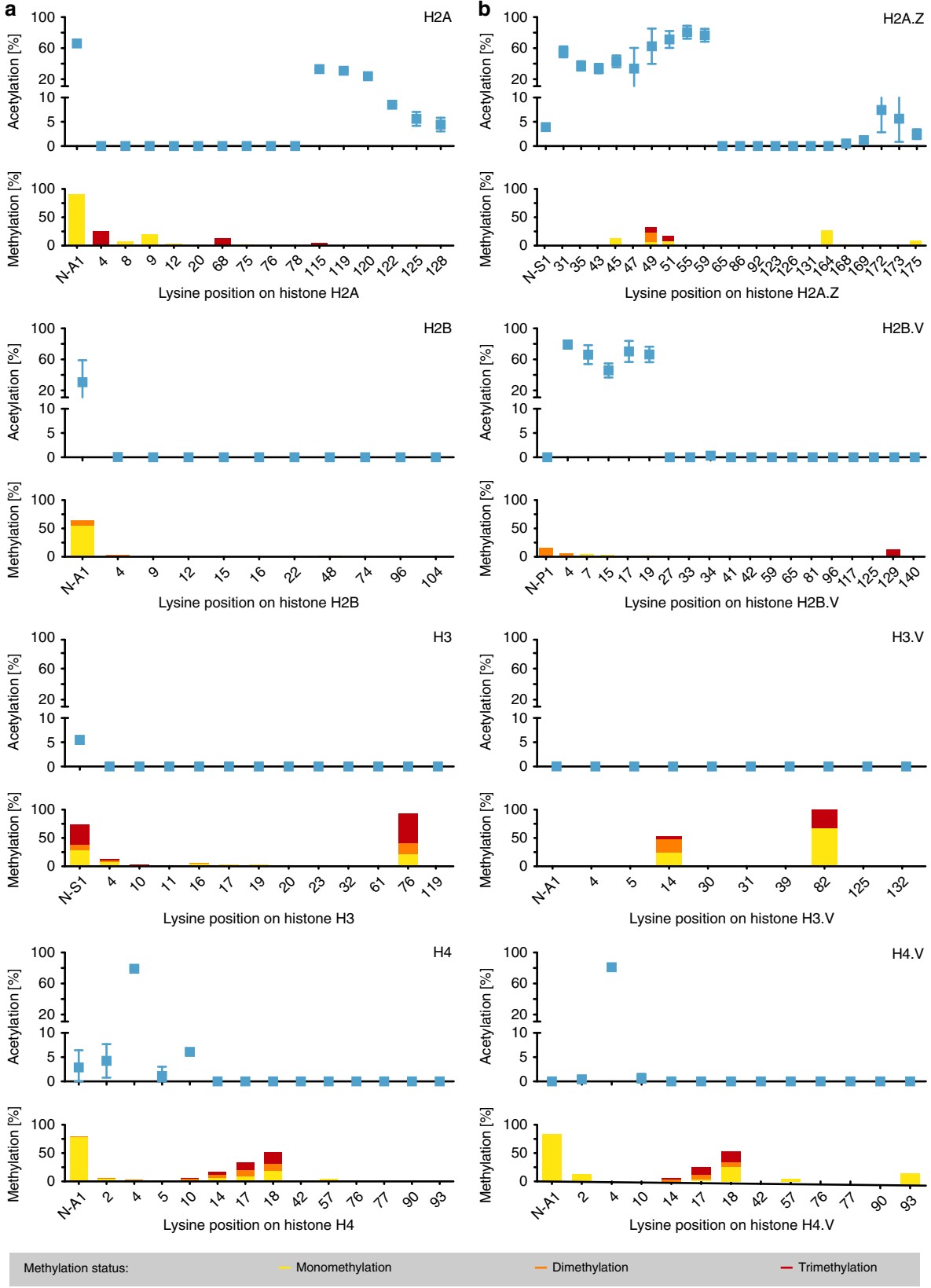

HAT1 and HAT2 transcripts resulted in impaired growth (Supplementary Fig. 8b, c and ref. [46]).

Following depletion of HAT1, we observed a significant reduction in acetylation of the N-terminal tails of H2A.Z, H2B.V as well as H4K2 (Fig. 5a and Supplementary Fig. 9a).

Interestingly, the other TSS-specific acetylation marks H4K5 and H4K10 were not affected.

Acetylome quantification of HAT2-depleted cells revealed a clearly distinct substrate pattern. Following depletion of HAT2 for 48 h, analysis of acid extracted histones from whole cell lysates

**Fig. 3 Quantification of histone acetyl and methyl marks. a** Lysine-specific acetylation levels (blue) are shown for core histones H2A, H2B, H3 and H4 and were determined by FIPQuant using histones isolated by acid extraction from wild type (WT) cells. The acetylation percentages [%] represent the averages of the median values from each of the independent experiments ($n = 3$) determined by FIPQuant. Error bars indicate standard deviations. Lysine-specific mono- (yellow), di- (orange) and/or tri- (red) methylation levels are shown for each core histone and were determined using spectra counting (lower panels). Stacked bars represent the average percentages of identified mono-, di- and trimethylated spectra from each of the independent experiments ($n = 3$). Supplementary Fig. 6a shows the data for each individual replicate. **b** Lysine-specific acetylation levels are shown for histone variants H2A.Z, H2B.V, H3.V, and H4.V (blue) and were determined by FIPQuant using whole histone extracts from WT cells for analysis of H3.V and H4.V or from immunoprecipitation of TSS-nucleosomes for analysis of H2A.Z and H2B.V. The acetylation percentages represent the averages of the median values from each of the independent experiments (H2A.Z and H2B.V: $n = 7$; H3.V and H4.V: $n = 3$) determined by FIPQuant. Error bars indicate standard deviations. Lysine-specific mono- (yellow), di- (orange) and/or tri- (red) methylation levels are shown for each histone variant and were determined using spectra counting (lower panels). Stacked bars represent the average percentages of identified mono-, di- and trimethylated spectra from each of the independent experiments (H2A.Z and H2B.V: $n = 7$; H3.V and H4.V: $n = 3$). Supplementary Fig. 6b shows data for each individual replicate. Source data are provided as a Source Data file.

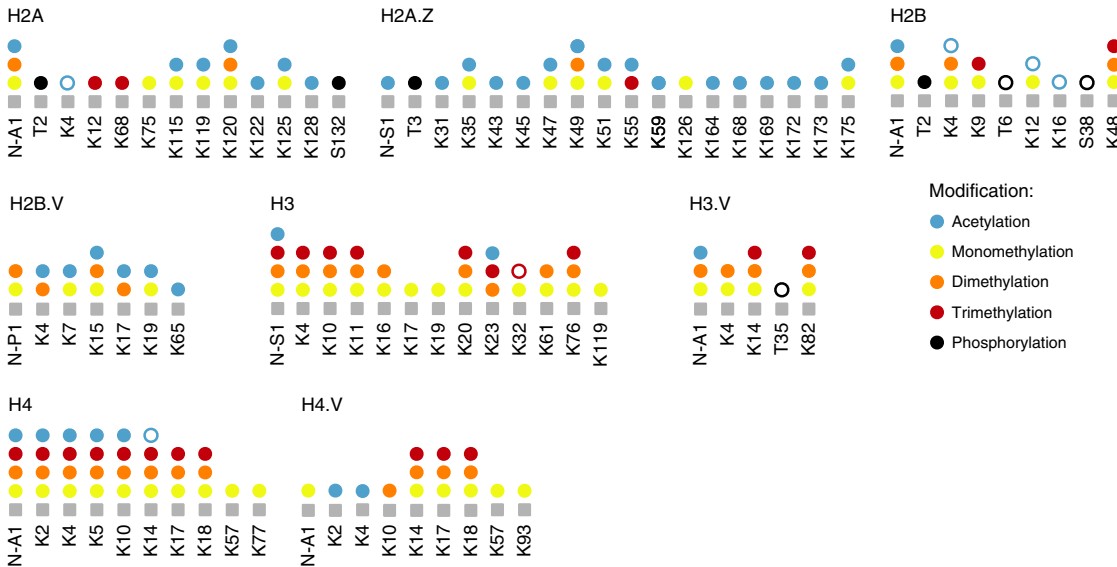

**Fig. 4 Summary of all acetylation, methylation and phosphorylation marks identified on _T. brucei_ histones.** PTMs detected in this study using acid extraction of histones from mononucleosomes or locus-specific histone isolation are indicated by solid circles. PTMs detected previously[22-25] but not in our study are indicated by empty circles. Supplementary Data 1 contains information on how often a specific PTM was detected and Supplementary Fig. 7 spectra of the newly identified phosphor marks. Source data are provided as a Source Data file.

indicated that all three TSS-specific acetyl marks (H4K2, H4K5 and H4K10) decreased to levels close to zero (Fig. 5b and Supplementary Fig. 9b). The impact on H2A.Z and H2B.V acetylation was much less pronounced than following depletion of HAT1 (Fig. 5). Interestingly, when we analyzed H4 co-immunoprecipitated with H2A.Z following HAT2 depletion, we still detected high levels of the TSS-specific marks H4K2ac and H4K10ac (Supplementary Fig. 10). Since much lower amounts of H4 co-immunoprecipitated with H2A.Z following HAT2 depletion, we did not obtain mass spectrometry coverage of the entire tail and thus could not evaluate H4K5ac levels. Nevertheless, combined, the decrease in the total amount of isolated H2A.Z/H4-containing nucleosomes following HAT2 depletion and the high abundance of H4 acetyl marks on the remaining H2A.Z/H4 nucleosomes suggest that H4 acetyl marks are required for the formation of H2A.Z-containing nucleosomes. Acetyl marks located on the other histones were not affected by depletion of HAT1 or HAT2 (Supplementary Fig. 11).

In summary, our data suggest that HAT1 and HAT2 are responsible for a distinct set of TSS-specific acetyl marks. HAT1 is primarily responsible for the acetyl marks on the histone variants H2A.Z and H2B.V while HAT2 acetylates the three TSS-specific acetyl marks on H4 (K2, K5, and K10).

**Depletion of HAT2 leads to a reduced deposition of H2A.Z.** Our observation that the subset of H2A.Z-containing nucleosomes still contains high levels of H4 acetylation even when HAT2 levels are low and TSS-specific H4 acetylation marks are reduced in total histone extracts, points to a role of H4 acetyl marks in H2A.Z deposition. To investigate this link in more detail, we depleted HAT1 or HAT2 using RNAi to reduce the acetyl levels on H2A.Z and H2B.V or on H4, respectively, and evaluated the effect on H2A.Z deposition.

Using MNase-ChIP-seq, we determined the distribution of H2A.Z in wild type and in HAT2 RNAi cells following 0 h ($n =$ 4), 24 h ($n = 2$), 48 h ($n = 4$) and 72 h ($n = 2$) of RNAi induction. These data revealed a clear link between HAT2 depletion and H2A.Z deposition. While the time scale of the effect varied among replicates, all replicates revealed a loss of TSS-specific H2A.Z deposition (Fig. 6a and Supplementary Fig. 12). A much weaker effect was observed following HAT1 depletion (Fig. 6b and Supplementary Fig. 13).

Since standard ChIP-seq assays do not yield information on the absolute amount of deposited H2A.Z, the change in H2A.Z distribution following HAT2 depletion could be the result of less H2A.Z being deposited at TSSs or increased deposition of H2A.Z at non-TSSs. To differentiate between these possibilities,

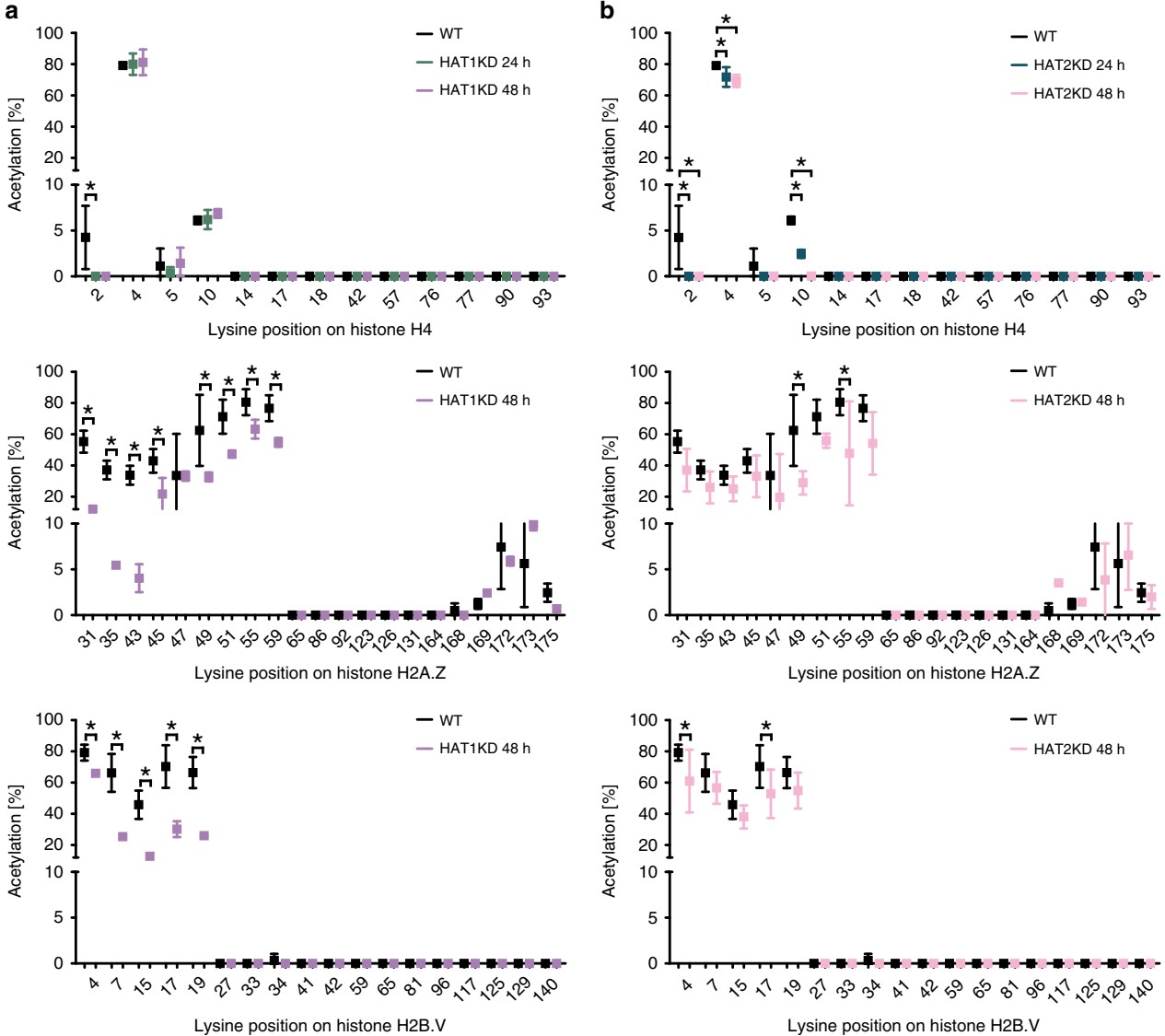

**Fig. 5 HAT1 and HAT2 acetylate histones at TSSs. a** Levels of lysine-specific acetylation for H4, H2A.Z and H2B.V are shown for WT cells and after depletion of HAT1. H4 acetyl marks were quantified by FIPQuant using histones extracted from WT cells (black, $n = 3$) and from 2T1 cells, which were depleted of HAT1 for 24 h (green, $n = 3$) or 48 h (purple, $n = 3$). H2A.Z and H2B.V acetyl marks were quantified using histones from immunoprecipitated TSS-nucleosomes (black; $n = 7$) and 2T1 cells, depleted for HAT1 for 48 h (purple; $n = 3$). The acetylation percentages [%] represent the averages of the median values from each of the independent experiments determined by FIPQuant. Error bars indicate standard deviations. Supplementary Fig. 9a shows the data for each replicate. **b** Levels of lysine-specific acetylation for H4, H2A.Z and H2B.V are shown for WT cells and after depletion of HAT2. H4 acetyl marks were quantified by FIPQuant using histones extracted from WT cells (black; $n = 3$) and from 2T1 cells, which were depleted of HAT2 for 24 h (blue; $n = 3$) or 48 h (rose; $n = 3$). H2A.Z and H2B.V acetyl marks were quantified using histones from immunoprecipitated TSS-nucleosomes of WT cells (black; $n = 7$) and 2T1 cells, depleted of HAT2 for 48 h (rose; $n = 2$). The acetylation percentages represent the averages of the median values from each of the independent experiments determined by FIPQuant. Error bars indicate standard deviations. Supplementary Fig. 9b shows the data for each replicate. Source data are provided as a Source Data file. Multiple t-tests between the different conditions were performed and individual p-values for each lysine position were computed using the two-way ANOVA approach. The statistical significance was determined using the Holm-Sidak method[77,78] and 'statistically significant' adjusted p-values ($p_{adj} < 0.05$) were marked with asterisks (exact p-values are listed in Supplementary Data 7).

we determined the amount of insoluble (chromatin-incorporated) H2A.Z by western blot analysis, using chromatin-bound H3 as a reference. A loss of deposition should lead to a decrease of the H2A.Z/H3 ratio in the insoluble fraction, while a loss of targeting, leading to a deposition across the PTUs, should not affect the H2A.Z/H3 ratio. Following depletion of HAT2, the western blot data indicate a strong reduction in the H2A.Z/H3 ratio to ~30% of wild-type levels (Fig. 6c and Supplementary Data 3).

Taken together our data suggest that following depletion of HAT2, acetylation of H4 at TSSs is reduced and this leads to a

decrease in H2A.Z deposition at TSSs. In contrast, depletion of HAT1, which resulted in reduced acetylation levels at H2A.Z and H2B.V, only minimally affected H2A.Z deposition.

**HAT2 depletion affects the site of transcription initiation.** Given the absence of canonical promoter motifs at RNA Pol II TSSs in *T. brucei*, it has been hypothesized that RNA Pol II transcription initiation is influenced by the extent of chromatin compaction, possibly induced by the presence of histone acetyl

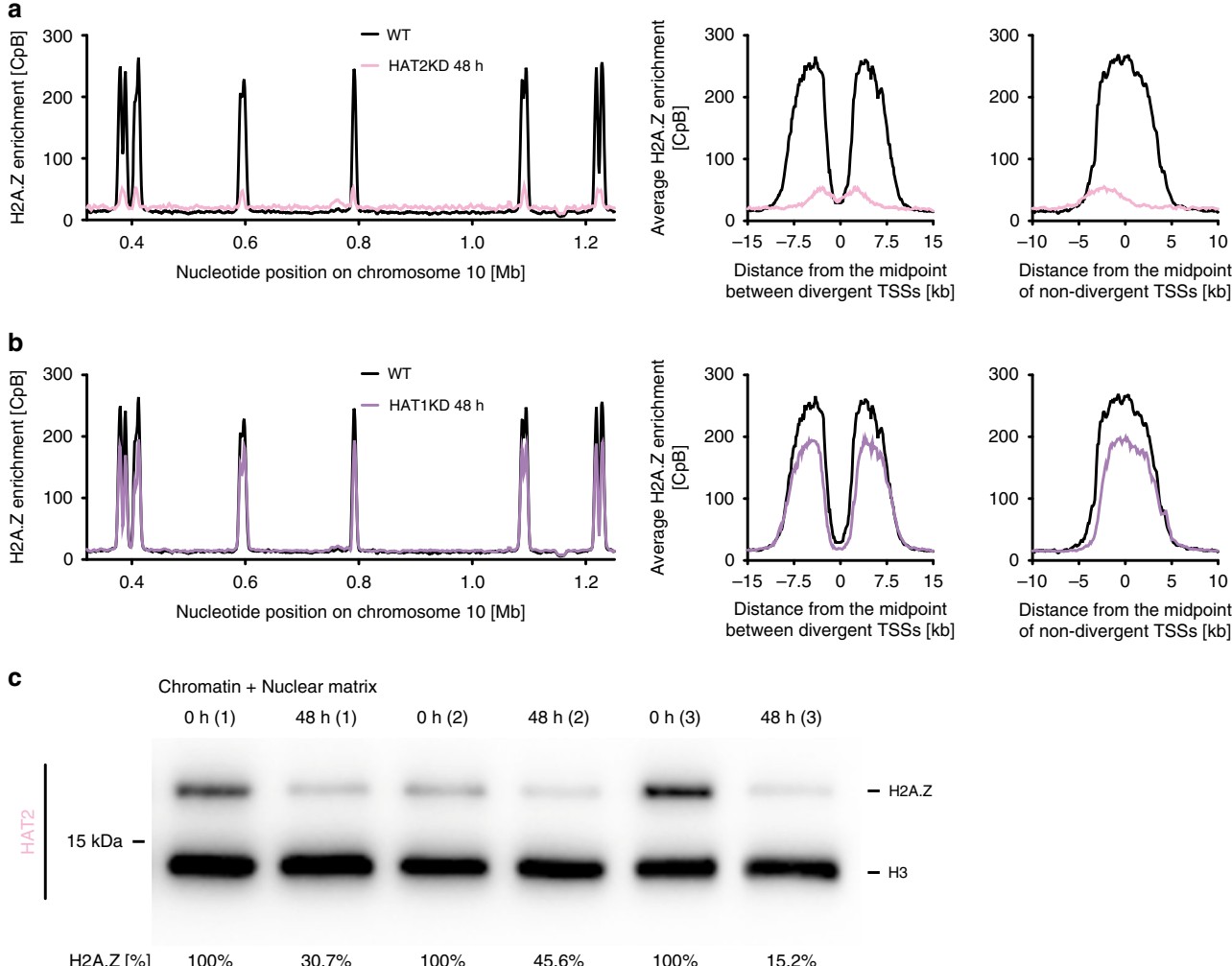

**Fig. 6 HAT2 depletion leads to reduced H2A.Z deposition. a** MNase-ChIP-seq data of histone variant H2A.Z before (black) and after 48 h HAT2 depletion (rose) are shown across a representative region of chromosome 10 (left panel) and averaged across divergent ($n = 37$) and non-divergent ($n = 49$) TSSs (right panel). The MNase-ChIP-seq data are normalized to the total number of reads and plotted as counts per billion reads [CpB]. Data from replicate 2 are shown, for all replicates see Supplementary Fig. 12. **b** MNase-ChIP-seq data of histone variant H2A.Z before (black) and after 48 h HAT1 depletion (purple) are shown across a representative region of chromosome 10 (left panel) and averaged across divergent ($n = 37$) and non-divergent ($n = 49$) TSSs (right panel). The ChIP-seq data are normalized to the total number of reads and plotted as counts per billion reads [CpB]. Data from replicate 1 are shown, for all replicates see Supplementary Fig. 13. **c** Western blot of chromatin-associated proteins extracted from 2T1 cells, which were depleted of HAT2 ($n = 3$) for 0 h or 48 h. Loaded are the insoluble fractions, containing chromatin-bound and nuclear matrix material. H2A.Z [%] refers to the H2A.Z/H3 ratio. The H2A.Z/H3 ratio at 0 h HAT2-depletion was set to 100%. H2A.Z/H3 ratios were calculated by quantifying the H2A.Z and H3 signal over the background for each lane signal using ImageJ (see Supplementary Data 3). Source data are provided as a Source Data file.

marks and histone variants[27,50,51]. While our current findings indicate that acetylation of H4 promotes the targeted deposition of H2A.Z, in vitro studies using recombinant histones have indicated that only acetylated H2A.Z (not unmodified H2A.Z) causes nucleosome destabilization[52]. Given our ability to independently reduce TSS-associated H4 acetylation and H2A.Z acetylation marks, we attempted to disentangle the functions of these marks and to test the hypothesis that H4 acetylation plays a role in defining the sites of transcription initiation whereas H2A.Z acetylation is mainly required for active transcription to occur.

To evaluate the effect of reduced H4 acetylation on RNA Pol II transcription, we compared the genome-wide distribution of DNA-associated RNA Pol II and RNA transcript levels between wild type and HAT2-depleted cells. If H4 acetylation influences RNA Pol II recruitment to TSSs, a reduction in H4 acetylation at TSSs would correlate with a change in the sites of RNA Pol II transcription initiation. To determine the genome-wide distribution of RNA Pol II enrichment by ChIP-seq, we used a cell line expressing TY1-tagged RPB9, a DNA-binding component present only in the RNA Pol II complex[53]. As described previously, RNA Pol II was strongly enriched at the 5′-end of TSSs[51]. Following depletion of HAT2 for 48 h, the peak of RNA Pol II enrichment increased in width, reaching further upstream than in wild-type cells (Fig. 7a and Supplementary Fig. 14a). In agreement with this observation, our RNA-seq analysis revealed a marked upstream shift in transcription initiation (Fig. 7b), with only minor changes in transcript levels detected across the PTUs (Fig. 7b and Supplementary Fig. 14b).

These results indicate that a reduction in TSS-associated H4 acetylation, resulting in a loss of H2A.Z deposition at TSSs, affects the site of transcription initiation but not transcription itself. Interestingly, analysis of data from previously published assay for transposase accessible chromatin with high-throughput sequencing (ATAC-seq)[31], indicates that the DNA just upstream of the

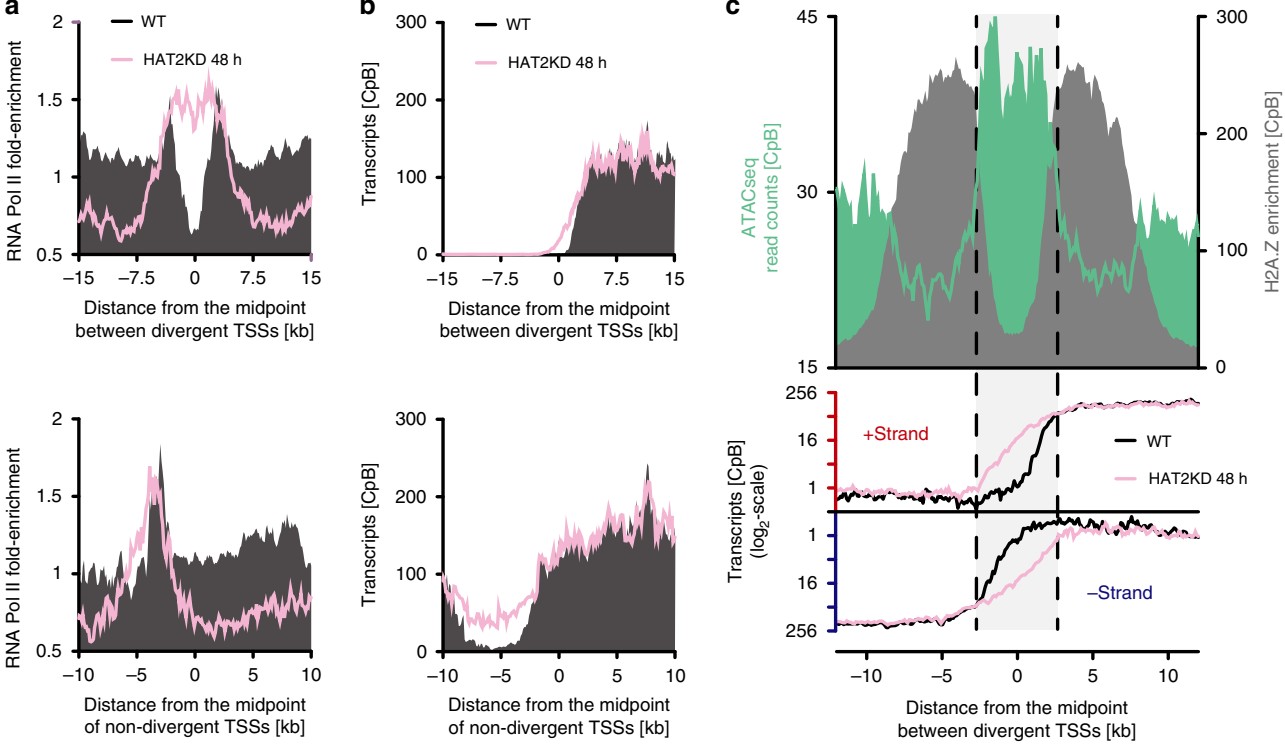

**Fig. 7 Depletion of HAT2 affects sites of transcription initiation. a** ChIP-seq data of RNA Pol II before (black) and after 48 h HAT2 depletion (rose) are averaged across divergent TSSs (n = 37; upper panel) and non-divergent TSSs (n = 49; lower panel). The ChIP-seq data are normalized to input data. Data from replicate 1 are shown, for all replicates see Supplementary Fig. 14. **b** RNA-seq data showing transcripts derived from the +strand before (black) and after 48 h HAT2 depletion (rose) are averaged across divergent TSSs (n = 37; upper panel) and non-divergent TSSs (n = 49; lower panel). The data are normalized to a spike-in control to account for differences in total RNA levels per cell after HAT2 depletion and plotted as counts per billion reads [CpB]. Shown is the average from three RNA-seq experiments (for normalization factors see Supplementary Data 5). **c** The ATAC-seq data (green)[31] and MNase-ChIP-seq data of H2A.Z (gray) are derived from wild type (WT) cells and are averaged across divergent TSSs (n = 37; upper panel). The ATAC-seq and MNase-ChIP-seq data are normalized to the total number of reads plotted as counts per billion reads [CpB]. RNA-seq data showing transcripts derived from the +strand or the −strand before (black) and after 48 h HAT2 depletion (rose) are averaged across divergent TSSs (n = 37; lower panel). The data are normalized to a spike-in control to account for differences in total RNA levels per cell after HAT2 depletion and plotted as counts per billion reads [CpB]. Source data are provided as a Source Data file.

canonical TSSs and the sites of H2A.Z is more accessible than the DNA at the TSSs itself. Thus, following a reduction of H2A.Z deposition, transcription appears to shift to sites of increased DNA accessibility (Fig. 7c).

In contrast to depletion of HAT2, depletion of HAT1, leading to reduced levels of H2A.Z acetylation, had no large effect on RNA Pol II enrichment across TSSs (Fig. 8a and Supplementary Fig. 15a). However, following HAT1 depletion we observed a marked, ~10-fold reduction of chromatin-bound RNA Pol II (Fig. 8b and Supplementary Data 4) and strong, twofold decrease in total RNA levels (Fig. 8c and Supplementary Fig. 15b). These findings suggest that loss of H2A.Z acetylation affects transcription. To be able to compare absolute transcript levels among different experiments, samples were spiked with a set of 92 synthetic transcripts (ERCC) of known concentration (Supplementary Data 5).

Together, these results indicate that while both H4 acetylation and H2A.Z acetylation are important for RNA Pol II transcription, they have distinct biological functions. Depletion of HAT2, leading to reduced H4 acetylation, affected the site of RNA Pol II initiation while depletion of HAT1, leading to reduced H2A.Z acetylation, led to an overall reduction in transcript levels.

## Discussion
Histone acetylation and variant forms of H2A seem to be present in almost every organism that contains histones and appear to

have a highly conserved role in regulating RNA Pol II transcription. The goal of this study was to elucidate how the absence of histone acetyl marks affects the deposition of H2A.Z and RNA Pol II transcription in vivo.

Using the unicellular parasite *T. brucei* and employing quantitative mass-spectrometry, we found TSS-nucleosomes to be highly acetylated. In addition, and unlike what has been observed in other organisms[16], we found H4 and H2A.Z to be acetylated by two different HATs, namely HAT2 and HAT1.

Taking advantage of this characteristic, we separately depleted the two HATs, which allowed us to specifically reduce the levels of H4 or H2A.Z acetyl marks and to unravel their effect on H2A.Z deposition and RNA Pol II transcription. Since in yeast and mammals, both H4 and H2A.Z are acetylated by the same HAT, Esa1 and Tip60 respectively[54,55], it has not been possible to perform similar experiments in these organisms to disentangle the role of these two marks in vivo[16]. Thus, while our data suggest some overlap between the role of HAT1 and Esa1/Tip60, we observed clearly distinct functions for HAT1 and HAT2. Depletion of HAT2, leading to reduced levels of TSS-associated H4 acetylation, had a large effect on H2A.Z deposition and the site of transcription initiation. In contrast, depletion of HAT1, leading to reduced levels of H2A.Z and H2B.V acetylation, had only a small effect on H2A.Z deposition, yet, it resulted in a strong reduction of transcript levels. Thus, it is possible that loss of variant acetylation results in a stabilization of variant-containing

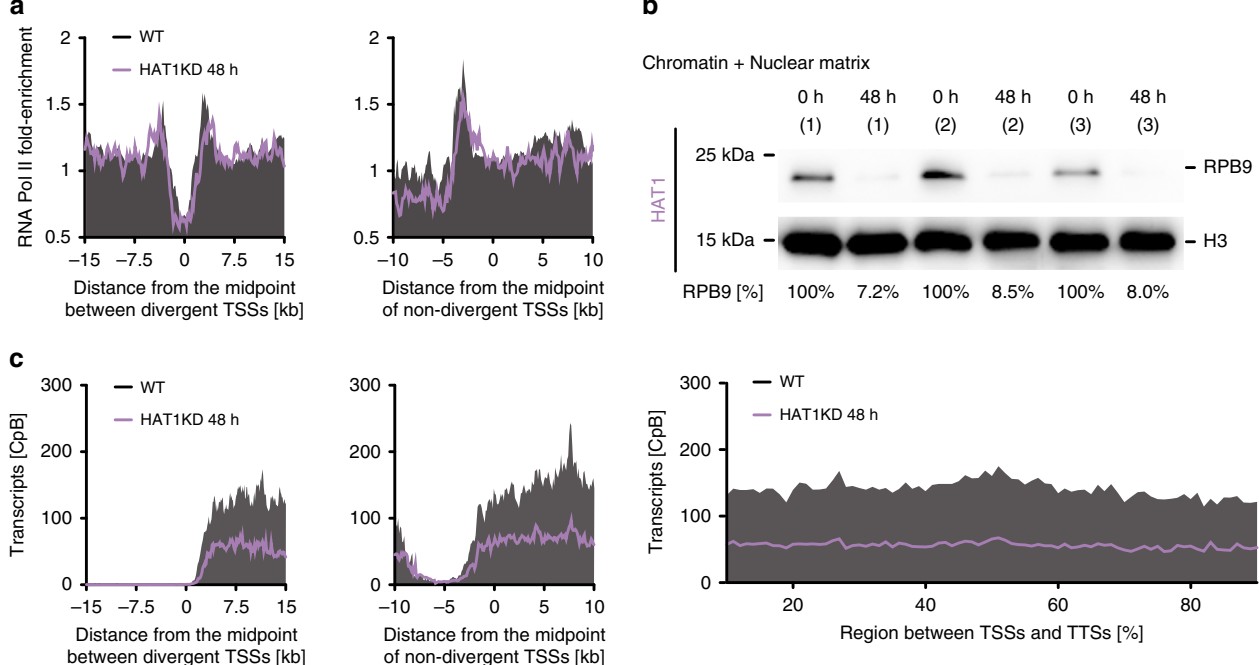

**Fig. 8 Depletion of HAT1 affects mRNA transcript levels. a** ChIP-seq data of RNA Pol II before (black) and after 48 h HAT1 depletion (purple) are averaged across divergent TSSs ($n = 37$; left panel) and non-divergent TSSs ($n = 49$; right panel). The ChIP-seq data are normalized to input data. Data from replicate 1 are shown, for all replicates see Supplementary Fig. 15. **b** Western blot of chromatin-associated proteins extracted from 2T1 cells, expressing TY1-tagged RPB9, before and after depletion of HAT1 ($n = 3$) for 48 h. Loaded are the insoluble fractions, containing chromatin-bound and nuclear matrix material. RPB9 [%] refers to the relative TY1-RPB9/H3 ratio. The ratio at 0 h HAT1-depletion was set to 100%. The TY1-RPB9/H3 ratios were calculated by quantifying the TY1-RPB9 and H3 signal over the background for each lane signal using ImageJ (Supplementary Data 4). **c** RNA-seq data showing transcripts derived from the +strand before (black) and after 48 h HAT1 depletion (purple) are averaged across divergent TSSs ($n = 37$; left panel), non-divergent TSSs ($n = 49$; right panel) and across regions between TSSs and TTSs on the +strand ($n = 109$; lower panel). The data are normalized to a spike-in control to account for differences in total RNA levels per cell after HAT1 depletion and plotted as counts per billion reads [CpB]. Shown is the average from three RNA-seq experiments (for normalization factors see Supplementary Data 5). Source data are provided as a Source Data file.

nucleosomes which in turn causes a defect in RNA Pol II recruitment and a reduction in DNA-associated RNA Pol II. An effect on RNA Pol II elongation seems unlikely, as this should have resulted in an increase in RNA Pol II levels at TSSs compared to the remaining part of the PTU.

Our mass-spectrometry analysis not only sheds light on the role of H4 and H2A.Z acetylation, it also enabled us to detect 157 PTMs, 126 of which had not been identified in *T. brucei* before. Furthermore, combining spectra counting to identify methyl marks, FIPQuant for the quantification of acetyl marks and the ability to specifically isolate TSS-nucleosomes, we were able to identify several acetyl and methyl marks enriched in TSS-nucleosomes. The large number of acetyl and methyl marks enriched at TSSs was surprising to us. *T. brucei* does not possess precise sites of RNA Pol II transcription initiation, but rather wide 'regions of transcription initiation'[51,56]. In addition, it is assumed that RNA Pol II transcription initiation is not regulated in *T. brucei* but that polymerases initiate at similar rates at all open and accessible TSSs[42]. These properties raise the question why such a large number of acetyl and methyl marks are enriched at TSSs and, assuming some of these marks serve as binding sites for acetyl- and methyl-binding proteins, why the parasite requires such a large number of proteins to be localized at TSSs. While we have not addressed these questions, we suspect that it is important for the parasite's survival that transcription initiates within clearly defined regions—regions marked by H2A.Z, H2B.V and the many newly identified TSS-specific acetyl and methyl marks. Given the presence of a highly efficient RNAi pathway in

*T. brucei*[57], transcription initiation upstream of native TSSs would, in many cases, lead to the generation of antisense RNA resulting in downregulation of some essential genes. In addition, it seems possible that even in the absence of gene-specific transcriptional regulation, a well-defined higher-order chromatin structure is required to ensure transcription from all PTUs and efficient processing of the transcripts.

In summary, our findings illustrate the importance of H2A.Z acetylation in active transcription suggesting an evolutionarily conserved link between histone acetylation, H2A.Z deposition and RNA Pol II transcription, even in organisms that lack canonical promoter motifs and precisely defined sites of transcription initiation.

## Methods

***Trypanosoma brucei* culture**. *Trypanosoma brucei* wild type (WT) and genetically modified strains derived from Lister 427 bloodstream-form MiTat 1.2 (clone 221a) strain, from a derivative "single marker" (SM) expressing a T7 polymerase and a Tet repressor[58] or from a derivative '2T1' strain expressing a Tet repressor and containing a puromycin-tagged ribosomal spacer as a landing pad for the transfection construct[59] were cultivated in HMI-11 medium (HMI-9 medium[60] without serum-plus) at 37 °C and 5% $CO_2$. When appropriate, the following drug concentrations were used: 2 µg ml$^{-1}$ G418 (neomycin), 5 µg ml$^{-1}$ hygromycin, 0.1 µg ml$^{-1}$ puromycin, 5 µg ml$^{-1}$ blasticidin, 2.5 µg ml$^{-1}$ phleomycin and 1 µg ml$^{-1}$ doxycycline. Growth rates were monitored for 96 h and cell densities were determined every 24 h. Transfections were performed using a Nucleofector (Amaxa) using an established protocol[61].

**Mononucleosome preparation for mass-spectrometry analyses**. To extract mononucleosomes from $2 \times 10^6$ bloodstream form trypanosomes following

previously established protocols[33], cells were harvested at 4 °C and 1800 × g for 10 min and washed using 40 ml of 1× TDB buffer (5 mM KCl, 80 mM NaCl, 1 mM MgSO₄*7H₂O, 20 mM Na₂HPO₄, 2 mM NaH₂PO4*2H₂O, 20 mM glucose). The cell pellet was resuspended in 1 ml of permeabilization buffer (100 mM KCl, 10 mM Tris 8.0, 25 mM EDTA 8.0, 1 mM DTT, 1.46 μM pepstatin A, 4.7 μM leupeptin, 1 mM PMSF and 1 mM TLCK, 50 mM sodium butyrate, 0.5 μM anarcadic acid) and centrifuged at 4 °C and 1800 × g for 10 min. The cell pellet was resuspended using 1 ml of permeabilization buffer. Digitonin was added to the cell suspension to a final concentration of 40 μM (stock solution: digitonin was dissolved to 4 mM in permeabilization buffer) and cell lysis was performed at RT for 15 min under gentle rotation. Following centrifugation, the pellet was washed three times in ice-cold isotonic buffer (100 mM KCl, 10 mM Tris 8.0, 10 mM CaCl₂, 5% Glycerol, 1 mM DTT, 1.46 μM pepstatin A, 4.7 μM leupeptin, 1 mM PMSF and 1 mM TLCK, 50 mM sodium butyrate, 0.5 μM anarcadic acid) for removing residual traces of EDTA. The cell pellet was resuspended in 1 ml of isotonic buffer and mononucleosomes were generated by micrococcal nuclease digestion (MNase; Sigma) using 2.5 U MNase and incubation for 15 min at 25 °C in a heating block. The reaction was stopped by adding 10 mM EDTA (pH 8.0). To the mononucleosomes-containing solution, 200 mM NaCl, 0.05% NP-40 were added. The reaction was incubated for 5 min on ice. Afterwards, the suspension was mixed by vortexing (five times, 10 s on, 30 s break). Following incubation on ice for 5 min without mixing, the mononucleosomes-containing suspension was cleared from other proteins by centrifugation at 4 °C and 11,000 × g for 10 min and the supernatant was transferred to a new 1.5 ml reaction tube.

**Acid extraction of histones from mononucleosomes**. To identify PTMs of all histones, irrespectively of their genomic location, histones were isolated from 1 × 10⁹ cells per replicate (in total: three replicates per approach, for an outline see Supplementary Fig. 2).

For the acid extraction of histones, 220 μl of 1 M H₂SO₄ were added to 1 ml mononucleosomes-containing supernatant extracted from 2 × 10⁶ bloodstream form trypanosomes according to a standard protocol for acid extraction[62]. Following an overnight incubation at 4 °C on a rotation wheel, the acid-insoluble proteins were removed by centrifugation at 4 °C and 16,000 × g for 10 min. The supernatant containing the free, acid-soluble histones was transferred into a new 1.5 ml reaction tube and the histones were concentrated using the StrataClean® resin (Agilent Technologies)[63]. Therefore, 20 μl StrataClean® resin was added and the suspension was incubated for 20 min at RT on a rotation wheel to allow binding of the histones to the resin. The resin was collected by centrifugation at RT and 16,000 × g for 1 min. To release the histones, the resin was incubated in 60 μl of 1× NuPAGE® LDS Sample Buffer (ThermoFisherScientific) supplemented with 50 mM DTT at 70 °C for 10 min and afterwards collected by centrifugation at RT and 16,000 × g for 1 min. The supernatant containing the histones from 2 × 10⁶ bloodstream form trypanosomes was transferred to a new 1.5 ml reaction tube. To reduce the histones for mass spectrometry analyses, the elution buffer was supplemented to contain 50 mM DTT.

**Locus-specific histone isolation**. Non-TSS-nucleosomes were purified in triplicates from cell line BFpFK8[33]. This cell line allows the inducible overexpression of TY1-tagged H2A from an rDNA locus[33]. For each replicate, mononucleosomes were isolated from 1 × 10⁹ cells as described above. Immunoprecipitation of non-TSS nucleosomes was performed using Dynabeads Protein G (Invitrogen) coupled to 10 μg monoclonal, purified BB2 mouse antibody[64] overnight (~14 h) at 4 °C. Bound material was washed three times with 1× PBS and eluted using 1× NuPAGE LDS Sample Buffer (ThermoFisherScientific) at 70 °C for 10 min. To reduce the histones for mass spectrometry analyses, the elution buffer was supplemented to contain 50 mM DTT.

To identify PTMs enriched at TSSs, TSS-nucleosomes were purified from cell line BFJEL41[33]. In this cell line both endogenous H2A.Z alleles are deleted and an ectopic copy of TY1-tagged H2A.Z is overexpressed. For each replicate, mononucleosomes were isolated from 1 × 10⁹ cells as described above. Immunoprecipitation of TSS-nucleosomes was performed overnight (~14 h) at 4 °C using Dynabeads Protein G (Invitrogen), which were coupled to 10 μg monoclonal, purified BB2 mouse antibody[64] and irreversibly crosslinked using dimethyl pimelidate (Sigma-Aldrich). Bound material was washed three times with 1× PBS and eluted using 1× NuPAGE LDS Sample Buffer (ThermoFisherScientific) at 70 °C for 10 min. To reduce the histones for mass spectrometry analyses, the elution buffer was supplemented to contain 50 mM DTT.

**Histone isolation for HAT1 and HAT2 target identification**. To identify the target sites of HAT1 and HAT2, we used cell lines allowing the inducible downregulation of HAT1 and HAT2 by RNAi (for cell line generation see Supplementary Information). Following depletion of HAT1 or HAT2, histones were isolated using the 'acid extraction of histones from mononucleosomes' protocol (see above). In addition, to enrich for H2A.Z and H2B.V-containing nucleosomes, we isolated TSS-nucleosomes similar to the approach described above. However, instead of using BB2 to pull out TY1-tagged histones, we used H2A.Z antibody[51] coupled to Dynabeads M-280 sheep anti-rabbit IgG (Invitrogen).

**Quantification of histone acetylation levels**. Purified histones were alkylated by adding iodacetamide (120 mM final concentration) followed by a 20 min incubation at RT. Next, histones were separated on NuPAGE Novex 4–12% Bis-Tris gels (Life Technologies) in MOPS buffer according to manufacturer's instructions and regions containing histones were excised from the gel. In-gel-acetylation was performed on dried gel pieces using a mixture of 10 μl of acetic anhydride-1,1'-¹³C₂ (Eurisotop) and 50 μl of 1 M sodium acetate-1-¹³C (Sigma-Aldrich; dissolved in H₂O water and adjusted to pH 7 using HCl). Mass spectrometry analysis and FIPQuant were performed as described before[32].

**Estimation of histone methylation levels**. Methylation levels were estimated by spectra counting. To this end the number of spectra for each modification type containing a specific modification site (e.g., di-methylated on K-4) were counted. The following modification forms were considered: unmodified, mono-methyl, di-methyl, tri-methyl as described before[32]. Spectra counting does not consider the differences in ionisation efficiencies between differentially modified peptides (e.g., mono- and di-methylated form of the same peptide) and is thus less accurate than FIPQuant.

**Identification of novel PTMs**. Data from FIPQuant were analyzed for additional PTMs with PEAKS studio X (Bioinformatics Solutions Inc., Canada). Raw data refinement was performed with the following settings: Merge Options: no merge, Precursor Options: corrected, Charge Options: no correction, Filter Options: no filter, Process: true, Default: true, Associate Chimera: yes. De novo sequencing and PEAKS database searching were performed with Parent Mass Error Tolerance set to 8 ppm. Fragment Mass Error Tolerance was set to 0.02 Da, and Enzyme was set to none. The following variable modifications have been used: Oxidation (M), Acetylation (K, protein N-terminal), Acetylation 13C1 (K, protein N-terminal), combined Methylation with Acetylation 13C1 (K, protein N-terminal), Tri-methylation (K, protein N-terminal), Dimethylation (K, protein N-terminal), Phosphorylation (STY). Carbamidomethylation (C) was set as fixed modification. A maximum of five variable PTMs were allowed per peptide. A custom database with all *T. brucei* histone sequences as well as common contaminants was used for database searching. Results were filtered to 0.5% PSM-FDR.

**MNase-ChIP sequencing**. MNase-ChIPs were performed using WT cells, the BFpFK8 cell line, which allows inducible overexpression of TY1-tagged H2A from an rDNA locus[33], and 2T1 cell lines, in which HAT1 or HAT2 can be inducibly depleted. In brief, 2 × 10⁸ cells were harvested, crosslinked in 1% formaldehyde, lysed using 200 μM digitonin (final concentration) and chromatin was fragmented using 1 U μl⁻¹ MNase (Sigma-Aldrich), for details see ref. [65]. Immunoprecipitation was performed using Dynabeads M-280 sheep anti-rabbit IgG (Invitrogen) coupled to 10 μg polyclonal affinity-purified H2A.Z rabbit antibody[51] or using Dynabeads Protein G (Invitrogen) coupled to 10 μg monoclonal, purified BB2 mouse antibody[64], overnight (~14 h) at 4 °C in the presence of 0.05% SDS (final concentration). Bound material was washed and eluted, and cross-links were reversed at 65 °C for ~11 h in the presence of 300 mM NaCl (final concentration). ChIP-seq libraries were constructed using 35 ng of immunoprecipitated or input DNA and library concentrations were determined in duplicates using Qubit dsDNA HS Assay Kit (Invitrogen, cat. no. Q32854) for Qubit 2.0 Fluorometer (Invitrogen, cat. no. Q32866). ChIP-seq libraries were quantified using the KAPA Library Quantification Kit (KAPA Biosystems, cat. no. KK4824) according to the manufacturer's instruction and sequenced in paired-end mode with 2 × 76 cycles using an Illumina NextSeq 500 platform.

**RNA Pol II ChIP sequencing**. RNA Pol II ChIPs were performed using 2T1 cell lines, in which both alleles of *RPB9*, a DNA-bound subunit of the RNA Pol II complex in *T. brucei*, contain an endogenous TY1-tag and HAT1 or HAT2 can be inducibly depleted. In brief, 3 × 10⁸ cells were harvested, crosslinked in 1% formaldehyde and lysed using 200 μM digitonin (final concentration). After centrifugation, the pellet was resuspended in 1000 μl NP-S buffer and sonicated using Covaris S220 focused-ultrasonication (Peak Intensity: 150, intensity: 5, Duty Factor: 10, Counts Per Burst: 200, Time: 10 min), for details see ref. [51]. After centrifugation, 60 μl of the supernatant was separated as input control. Immunoprecipitation of fragmented RPB9-bound DNA was performed using Dynabeads Protein G (Invitrogen), coupled to 10 μg monoclonal, purified BB2 mouse antibody[64], overnight (~14 h) at 4 °C in presence of 300 mM NaCl (final concentration). Bound material was washed and eluted, and cross-links were reversed at 65 °C for ~11 h in the presence of 300 mM NaCl (final concentration). ChIP-seq libraries were constructed using ~5 ng of immunoprecipitated or 35 ng input DNA, size-selected (150–350 bp) on a native PAGE gel and library concentrations were determined in duplicates using Qubit dsDNA HS Assay Kit (Invitrogen, cat. no. Q32854) for Qubit 2.0 Fluorometer (Invitrogen, cat. no. Q32866). ChIP-sequencing libraries were quantified using the KAPA Library Quantification Kit (KAPA Biosystems, cat. no. KK4824) according to the manufacturer's instruction and sequenced in paired-end mode with 2 × 76 cycles using an Illumina NextSeq 500 platform.

**Mapping, normalization, and visualization of ChIP-seq data.** ChIP-seq reads were mapped to *T. brucei* Lister 427 genome version 9 (HGAP3_Tb427v9; from Zenodo DOI 10.5281/zenodo.823671 [https://doi.org/10.5281/zenodo.823671][31], using Bowtie 2 *local*[66]. Following SAM to BAM conversion, the aligned reads were sorted and indexed using SAMtools version 1.8[67]. The number of reads was normalized per billion mapped reads and coverage files were generated in the wiggle format using COVERnant version 0.3.0 with the subcommand *ratio*[51]. For visualization, regions of interest were extracted from the wiggle files and the coverage was illustrated using GraphPad Prism version 7.0c.

The coverage for multiple regions was extracted and averaged using COVERnant *extract*. Meta-plots were generated by plotting the median without zeros of the output-matrices to the indicated locations and illustrated using GraphPad Prism version 7.0c.

**RNA sequencing.** RNA sequencing was performed in triplicates for WT cells and in 2T1 cells, in which HAT1 or HAT2 were inducibly depleted for 48 h. In brief, total RNA was extracted from $4.5 \times 10^7$ cells grown to a density of $0.9 \times 10^6$ cells ml$^{-1}$ using the NucleoSpin RNA kit (Macherey-Nagel; cat. no. 740955.10). 3.8 μl of 1 M RNAse-free DDT (Sigma-Aldrich; cat. no. 10197777001) and 1 μl of 1:10 Ambion ERCC RNA Spike-In Mix (ThermoFisherScientific; cat. no. 4456739) were added to the cell lysis buffer. Ribosomal RNA was depleted from 2 μg of total RNA, cDNA was synthesized using the NEBNext Ultra Directional RNA Library Prep Kit for Illumina (New England Biolabs; cat. no. E7420), strand-specific RNA-seq libraries were generated, for details see ref. [68]. Library concentrations were determined in duplicates using Qubit dsDNA HS Assay Kit (Invitrogen, cat. no. Q32854) for Qubit 2.0 Fluorometer (Invitrogen, cat. no. Q32866). Strand-specific RNA-sequencing libraries were quantified using the KAPA Library Quantification Kit (KAPA Biosystems, cat. no. KK4824) according to the manufacturer's instruction and sequenced in paired-end mode on an Illumina NextSeq 500 sequencer $2 \times 76$ cycles.

**Mapping, normalization, and visualization of RNA-seq data.** Adapter sequences were removed from the raw sequencing files using Cutadapt[69] and the RNA-sequencing datasets were mapped to a hybrid genome, which contains the *T. brucei* Lister 427 genome version 9 (HGAP3_Tb427v9; from Zenodo DOI 10.5281/zenodo.823671 [https://doi.org/10.5281/zenodo.823671]) and the sequences of the 92 ERCC spike-in transcripts, using BWA-mem[67]. Following SAM to BAM conversion, the aligned reads were sorted and indexed using SAMtools version 1.8 and unmapped, PCR or optical duplicates, not primary aligned and supplementary aligned reads were removed from the alignment files (SAM flag: 3332)[67].

Using BAM files, reads per *T.brucei* CDS were counted using HGAP3_Tb427v9.gff as annotation file and reads per ERCC transcripts were counted using ERCC92.gff using the GenomicAlignments package[70,71] in R[72]. Differential gene expression analysis was conducted using the DESeq2 package from R/Bioconductor, normalizing the counts per *T. brucei* gene to the ERCC spike-in counts[73]. Features with an adjusted *P* value (calculated based on Wald test and adjusted for multiple testing using the procedure of Benjamini and Hochberg) below 0.1 were considered as differentially expressed[74]. To analyze the depletion efficiency for HAT1 and HAT2, the CDS information of HAT1 and HAT2 in the annotation file was split into the RNAi target, the upstream target and the downstream target region.

Additionally, the number of reads was normalized to reciprocal ERCC factors, calculated using the ERCC factors from the DESeq2 analysis, to generate coverage files in the wiggle format using COVERnant version 0.3.0 with the subcommand *ratio* including the additional parameters *−factor_numerator* and *−factor_denominator*. For visualization, regions of interests were extracted from the wiggle files and the coverage was illustrated using GraphPad Prism version 7.0c. Meta-plots were generated by plotting the median without zeros of the output-matrices to the indicated locations and illustrated using GraphPad Prism version 7.0c.

**Extraction of chromatin-associated proteins.** To determine the amount of deposited H2A.Z or chromatin-associated RNA Pol II in comparison to histone H3 following HAT depletion, we determined the amount of insoluble (chromatin-incorporated) H2A.Z or TY1-RPB9 by western blot analysis, using chromatin-bound H3 as control. The protocol for extraction of chromatin-associated proteins was kindly provided by Joana Faria (University of Dundee). Analysis of chromatin-associated proteins was performed in 2T1 cells, in which HAT2 was depleted for 48 h, and in 2T1 cells, in which both endogenous *RPB9* contain a N-terminal TY1-tag and in which HAT1 was depleted for 48 h. For each sample $4 \times 10^6$ cells were harvested by centrifugation at 4 °C and $1500 \times g$ for 10 min and washed in 1 ml of $1 \times$ TDB (5 mM KCl, 80 mM NaCl, 1 mM MgSO$_4$, 20 mM Na$_2$HPO$_4$, 2 mM NaH$_2$PO$_4$, 20 mM glucose pH 7.4) followed by centrifugation at 4 °C and $1500 \times g$ for 10 min. The cell pellet was resuspended in 20 μl of freshly prepared CSK-buffer (100 mM NaCl, 0.1% Triton X-100, 300 mM Sucrose, 1 mM MgCl$_2$, 1 mM EGTA, 10 mM PIPES (pH 6.8; with NaOH) supplemented to contain 1 mM Pepstatin A, 5 mM Leupeptin, 100 mM PMSF and 100 mM TLCK (final concentration)) and incubated on ice for 10 min. To separate the soluble (cytosol + nucleosoluble) from the insoluble (chromatin + nuclear matrix) fraction, the cell suspension was centrifuged at $2550 \times g$ and 4 °C

for 5 min. The supernatant containing the soluble fraction was separated and the pellet was washed using 20 μl of CSK-buffer at $2550 \times g$ and 4 °C for 5 min. The supernatant from the washing step was separated from the pellet, which contains the insoluble fraction. To 23 μl of soluble fraction, 7 μl of $4 \times$ sample buffer ($4 \times$ Laemmli buffer supplemented with 2.5% β-Mercaptoethanol, 1 mM Pepstatin A, 5 mM Leupeptin, 100 mM PMSF and 100 mM TLCK (final concentration)) were added and the pellet containing the insoluble fraction was resuspended in 30 μl of $1 \times$ sample buffer. Proteins were denatured at 90 °C for 10 min. The insoluble fraction of $2 \times 10^6$ cells was analyzed by western blotting.

For detection of H2A.Z and H3 the polyclonal affinity-purified H2A.Z rabbit antibody (1:750; stock concentration 1 μg μl$^{-1}$) and the polyclonal H3 rabbit antiserum (1:20,000) were used[51,75]. For detection of TY1-RPB9 the monoclonal, purified BB2 mouse antibody[64] (1:750; stock concentration 1 μg μl$^{-1}$) was used.

**Cell line generation and additional methods.** For details on *T. brucei* manipulation, plasmids, additional methods and sequencing datasets, see Supplementary Information and Supplementary Data 6.

**Reporting summary.** Further information on research design is available in the Nature Research Reporting Summary linked to this article.

## Data availability
All sequencing data and coverage tracks generated for this publication have been deposited in the Gene Expression Omnibus (GEO) and will be accessible through the accession number GSE145812. Mass spectrometry proteomics data generated for this publication have been deposited in the ProteomeXchange Consortium via the PRIDE[76] partner repository and will be accessible through the accession number PXD014452. The source data underlying Figs. 1a, b, 2b, 3a, b, 4, 5a, b, 6a–c, 7a–c, 8a–c, and Supplementary Figs. 1a–c, 10a, b, 11, 12, 13, 14a, b and 15a, b are provided as a Source Data file.

## Code availability
The R scripts for performing FIPQuant can be downloaded from github [https://github.com/JTVanselow/Patchwork_AcK_Quant] and are deposited at Zenodo DOI 10.5281/zenodo.3662776 [https://doi.org/10.5281/zenodo.3662776]. Documentation to reproduce the data analysis is provided. Workflows and custom-made scripts are deposited at Zenodo DOI 10.5281/zenodo.3662776 [https://doi.org/10.5281/zenodo.3662776].

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

## Acknowledgements

We thank all current and former members of the Siegel and Janzen laboratories for valuable discussions and for assistance with experiments. We thank Joana Correia-Faria for sharing protocols for the isolation of the chromatin-associated protein fractions. We thank Felix Müller-Planitz, Stan Gorski, Kirsty McWilliam, and Raúl Cosentino for critically reading the manuscript. We thank Konrad U. Förstner and Raúl O. Cosentino for suggestions regarding data analysis and the Core Unit Systems Medicine of the University of Würzburg for the high-throughput sequencing. We acknowledge the support and resources from the Bioinformatics Core Facility at the Biomedical Center Munich. We thank Beate Vogt and Christiane Winkler for their help in sample preparation for FIPQuant. This work was funded by the Young Investigator Program of the Research Center for Infectious Diseases (ZINF) at the University of Würzburg, Germany, a grant from the German Research Foundation (SI 1610/2-1) and an ERC Starting Grant (3D_Tryps 715466).

## Author contributions

The study was conceptualized by A.J.K. and T.N.S. Experimental work was carried out by A.J.K. FIPQuant was performed by S.L., J.T.V., and A.S. Sequencing data analysis was performed by A.J.K. B.G.B. performed the differential gene expression analyses and wrote the *meta-plot* subcommand for COVERnant. The manuscript was written by A.J.K. and T.N.S.

## Competing interests

The authors declare no competing interests.
