## [Peer Review File · Nature Communications]

Reviewers' Comments:

Reviewer #1:

Remarks to the Author:

Kraus et al. use ChIP followed by mass spec to identify 157 distinct post-translational modifications of histones in *Trypanosoma brucei*, some 58 of which are enriched at transcriptional start sites (TSSs). They take advantage of histone variants and the unusual extended structure of TSSs in trypanosomes to separate TSS nucleosomes from others, and quantify acetylation and methylation levels at specific histone residues in TSS and non-TSS nucleosomes. They identify two MYST family histone acetyl transferases, HAT1 and HAT2, that have differential effects on acetylation of H2A.Z and H4, respectively. Using RNAi knockdown of these HATs, they show that acetylation of H4 is necessary for efficient H2A.Z loading and acetylation of H2A.Z is necessary for efficient transcription. This interesting paper should appeal to researchers interested in transcription and gene regulation, acetylation, histone variants, and chromatin evolution.

I have only minor comments:

p. 3 - "others such as H3.V, CENP-A, or H2A.Z are widely conserved in evolution. Thus far, H2A.Z remains the only histone variant identified in all eukaryotes". H3.V is unique to trypanosomes, did the authors mean H3.3? H2A.Z is absent in sequenced metamonads (*Giardia* and *Trichomonas*) See. Dalmaso et al 2011 (PMID: 21622164)

p.5 - "The mutually exclusive distribution of the two histones points to an absence of heterotypic H2A.Z/H2A nucleosomes". Is this strictly accurate? In figure 1, TY-H2A fold-enrichment, though clearly anti-correlated with H2A.Z enrichment, does not go to zero at TSSs. Did any H2A turn up in mass spec of H2A.Z containing nucleosomes, or vice versa?

Figure 2.- Although the legend is clear enough, the Y axis label does not mention methylation. I would suggest letting the label be on two lines if needed so that it can read "Methylation level (%)
acetylation level (%)" or similar parallel construction, rather than "Fraction [%] Acetylation degree in %".

p. 7 - In the paragraph beginning "In addition to H4, H2A.Z and H2B.V...", "Supplementary Figure 4 and b" should be "Supplementary Figures 4a and 4b". In the next sentence "We observed a similar pattern for H2B.V and H2B." The pattern for H2B is not similar. Please adjust the text. In the next paragraph, the authors say "For Histone H4 methylation, we did not observe any notable differences between TSSs and non-TSSs." Is monomethylation of H4K2 in TSSs not a notable difference? The difference appears to be 30-40% vs 0%.

p. 12 and Figure 6b - "No such effect was observed following HAT1 depletion." Though clearly much less dramatic than HAT2 depletion, H2A.Z deposition or occupation does appear to be slightly reduced after HAT1 depletion. (Is figure 6b normalized to total reads?) Similarly, two paragraphs later "In contrast, depletion of HAT1 ... does not affect H2A.Z deposition" perhaps should be "In contrast, depletion of HAT1 ... only minimally affects H2A.Z deposition."

Discussion - Is there any rationale for why pol II moves upstream of the TSS regions in HAT2-depleted cells? What type of nucleosomes reside there normally, or after HAT2 depletion?

p. 15, top - "Both HAT1 and HAT2 may be evolutionary ancestors of the enzymes ... pointing to a fusion of both enzymes into one acetyltransferase during evolution." This seems rather odd. Is there evidence that NuA4 or p400 contain two acetyltransferase domains? It seems more likely that the substrate specificity of HAT1, HAT2, or some other HAT might change to be able to acetylate both H4 and H2A.Z, or that might be the ancestral condition, with specialization of substrates in trypanosomes. A phylogenetic tree of the of the various HATs in question seems like it might be able to confirm or dispute this model.

Supplementary Figure 2 - H4 is nearly invisible. Is it possible to increase the contrast or intensity to make it more visible?

Supplementary Figure 13 (and others) - The colors used to depict 24 hr and 48 hr knockdowns are

too similar to each other to be easily distinguished. This is important in part b, where they give very different profiles. In the legend "The data is normalized" should be "The data are normalized" (occurs twice).

Reviewer #2:

Remarks to the Author:

In this manuscript, Kraus and colleagues characterise and compare the chromatin environment at transcription start sites (TSS) and within polycistronic transcription units (PTUs) in *Trypanosoma brucei*. They then use these new data to explore the effects of RNAi against two acetyltransferases, HAT1 and HAT2, on patterns of acetylation of histone H4 and H2A variants, as well as localisation of the histones and RNA Polymerase II (Pol II), and on mRNA expression. The abstract (and later text) states that the outcome of this investigation is that 'specifically reducing H4 or H2A.Z acetylation levels enabled us to reveal distinct roles for these modifications with regard to H2A.Z deposition and RNA transcription'. This would be important, as it would represent a clear case of resolving cause and effect in the complex orchestration of chromatin deposition and gene expression control. However, I suggest that the statement is somewhat inaccurate and clarification in several areas is needed to allow us to reach such a clear conclusion.

Main issues to be addressed.

1. The chromatin experiments that serve as the foundation for the subsequent RNAi analysis suffer from lack of detailed description, meaning the reader is left to take it on trust that everything is ok. Several aspects need to be resolved.

i). The key approach in this paper is to use TY-H2A.Z and TY-H2A immunoprecipitation to isolate nucleosomes from the TSSs and PTUs, respectively, based on the observation that the variant histone is enriched at TSSs while the core histone is depleted. The authors need to more clearly explain exactly how exactly the experiment was performed. In the results they merely state 'By taking advantage of the distinct genome-wide distributions of H2A.Z and H2A, we were able to enrich specifically TSS-nucleosomes (those containing H2A.Z) and non-TSS nucleosomes (those containing H2A)'.

It is very hard work to decipher how the 'histones were purified' in this approach, or what level of purity was seen. Please expand on experimental descriptions in the results and methods.

Equally, greater explanation is needed for what is meant by 'fraction [%]' and 'acetylation degree' in Fig2b. How was the methylation status determined (since Fig.2a only describes the assay used for identifying acetylation)? (The only explanation I could find is: 'For a semi-quantitative analysis of histone methyl marks, we relied on peptide counts'.)

How should we consider the two forms of quantification in terms of what they can tell us, since there is little overlap in the acetylation between the two datasets in Fig2b, but considerable overlap for methylation?

In Fig3, how were the analyses conducted, as the current description ('Lysine-specific acetylation degrees are shown for histone variants H2A.Z, H2B.V, H3.V and H4.V (blue) and were determined by FIPQuant using whole histone extracts from WT cells or from immunoprecipitation of TSS-nucleosomes') is inadequate: presumably it is impossible to use either TY-HA2 or Ty-H2A.Z to analyse H3.V and H3.V at TSSs, since they have only been reported at termination sites (see also below)?

Please state clearly how each dataset was generated.

ii). The wording of the text suggests that TY-H2A.Z and TY-H2A immunoprecipitations are very discriminatory between TSSs and PTUs and, indeed, locus-specific (ie TSS) chromatin isolation is a key claim of the paper overall. Is this true, and how do we know this?

From Fig.2 there appears to be quite considerable data overlap in the two datasets, so how have the authors critically tested the claimed specificity?

Were the same approaches applied for all the data in Fig.3? For instance, how were H3.V and H4.V examined, since there are found at termination sites and would not be expected to be recovered in either of the TY-H2A.Z or TY-H2A immunoprecipitations?

iii). What material was used for the acid extraction analysis shown in Fig.4? Is this the methods section entitled 'Extraction of chromatin-associated proteins' and, if so, in what way does this improve on the 'technical reasons these [previous] studies failed to obtain full coverage of core histone sequences'?

2. HAT2. The combined data in Figs 5, 6 and 7 is interpreted to suggest that RNAi leads to a specific loss of histone H4 acetylation on lysines 2, 5 (perhaps) and 10, leading to altered deposition of histone H2A.Z and RNA Pol II, with a small change in transcription initiation site but no clear loss of mRNA. This is a nice set of data, but it is complete?

One simpler explanation, not explored and less mechanistically clean, is that the RNAi has a dramatic effect on overall levels of H2A.Z, which could account for the ChIPseq and might explain the lack of specific Pol II localisation at the TSSs. Have they run western on whole cell extracts after RNAi?

Secondly, do we know that HAT2 RNAi has the stated, highly specific effect on histone H4, and that this directly leads only to the effects described on H2A.Z and Pol II? Did the authors run acetylation assays in the RNAi cells relative to WT to look at all histones?

3. HAT1. Here, the authors show that RNAi leads to widespread effects on acetylation of H4, H2A.Z and H2B.V, with associated loss of mRNA that cannot be explained by altered H2A.Z localisation. While it is true that this suggests a distinct response to HAT1 RNAi, the experiments do not provide a mechanistic explanation and, at least superficially, the findings seem comparable to data on Esa1 and Tip60 function in yeast and humans. Greater discussion of this overlap is warranted.

Moreover, why do they authors discount the possibility that HAT1 and HAT2 arose by gene duplication in trypanosomes to provide specialised functions? At the moment they seem to think all (or most) other eukaryotes have undergone an HAT1-2 gene fusion, which seems a more far-fetched explanation: 'Both HAT1 and HAT2 may be evolutionary ancestors of the enzymes responsible for H4 as well as for H2A.Z acetylation in other eukaryotes pointing to a fusion of both enzymes into one acetyltransferase during evolution'.

Reviewer #3:

Remarks to the Author:

The manuscript by Kraus et al. „Distinct roles for H4 and H2A.Z acetylation in RNA transcription within African trypanosomes“ describes the proteomic analysis of the Trypanosoma histone acetylation and the roles of distinct sites in the recruitment of histones to the transcription sites and the transcription

initiation, as well as the roles two histone acetyltransferases in these processes. The data would be of interest to many biologists in several fields. The manuscript is well written and the experimental design is good, the logic of the experiments is flawless in my opinion, however, it seems that the proteomics part is much stronger than the functional part. The proteomic analysis created the unique and comprehensive database of histone acetylations in *Trypanosoma brucei*, taking advantage of a unique chromatin structure of T-brucei and a relatively new method of site identification, FIPQuant. The impact of the two HATs is less solidly demonstrated. Starting with Figure 5, it becomes not 100% clear that the two HATs have indeed so distinct roles as the authors surmise. This reviewer would like to see the statistical analysis associated with the measurements in Figure 5 (with p-values associated with the differences) - this is a minor revision.

Response to Reviews

We would like to thank the reviewers for their constructive criticisms. We believe that the revised version of the manuscript addresses all points raised by the reviewers. All relevant changes have been highlighted in the manuscript.

Please find below our responses to the issues raised by the referees.

Referee #1:

I have only minor comments:

p. 3 - “others such as H3.V, CENP-A, or H2A.Z are widely conserved in evolution. Thus far, H2A.Z remains the only histone variant identified in all eukaryotes”. H3.V is unique to trypanosomes, did the authors mean H3.3?

We had used the term H3.V to refer to H3 variants in general. However, we understand that this is confusing and have replaced H3.V with H3.3 as suggested by the referee (page 3).

H2A.Z is absent in sequenced metamonads (*Giardia* and *Trichomonas*) See. Dalmasso et al 2011 (PMID: 21622164)

We thank the reviewer for the reference and added the information concerning H2A.Z to our introduction (page 3).

p.5 - “The mutually exclusive distribution of the two histones points to an absence of heterotypic H2A.Z/H2A nucleosomes”. Is this strictly accurate? In figure 1, TY-H2A fold-enrichment, though clearly anti-correlated with H2A.Z enrichment, does not go to zero at TSSs. Did any H2A turn up in mass spec of H2A.Z containing nucleosomes, or vice versa?

In addition to the ChIP-seq analysis of H2A.Z and H2A, we had performed co-IPs of H2A.Z and H2A-containing nucleosomes. By western blotting we could not detect any H2A in H2A.Z-containing nucleosomes and, just as published previously, we could not detect any H2A.Z in H2A-containing nucleosomes, pointing to very low levels of heterotypic nucleosomes. However, analysis of the co-immunoprecipitated histones by mass spectrometry revealed some H2A peptides in the H2A.Z IP and some H2A.Z peptides in the H2A IP. Mass spectrometry is obviously much more sensitive than western blotting, allowing the detection of minute levels of a protein. The H2A.Z detected in the H2A IP and the H2A detected in the H2A.Z could represent small amounts of heterotypic nucleosomes. Alternatively, they could be the results of low levels of di-nucleosomes in our preparation.

Nevertheless, we feel that our ability to quantify acetyl levels allows us to identify marks enriched at TSS-nucleosomes or non-TSS-nucleosomes even if H2A.Z and H2A distribution is not fully mutually exclusive.

We have added our co-IP results and removed the claim that distribution of H2A.Z and H2A is mutually exclusive. Instead we say:

“These findings point to very low levels of heterotypic H2A.Z/H2A nucleosomes [...] and should allow us to enrich for nucleosomes from TSSs (containing H2A.Z) and non-TSSs (containing H2A)” (page 5).

Regarding the question why H2A levels do not go to zero in our ChIP-seq analysis:

In Fig. 1a we are plotting the ratio of H2A IPed DNA to input DNA, thus, we would not expect the values to go to zero. In the previous version we had incorrectly used a scale going to 0 for the H2A.Z ratio. This has been corrected in the revised manuscript. Now, we are only using scales going to 0 when we show counts per billion (CPB), not when showing ratios.

Figure 2.- Although the legend is clear enough, the Y axis label does not mention methylation. I would suggest letting the label be on two lines if needed so that it can read “Methylation level (%) acetylation level (%)” or similar parallel construction, rather than “Fraction [%] Acetylation degree in %”.

We adjusted the labels of the y-axis in Fig. 2 and 3 and in the Supplementary Fig. 4 and 6 as suggested by the reviewer.

p. 7 – In the paragraph beginning “In addition to H4, H2A.Z and H2B.V...”, “Supplementary Figure 4 and b” should be “Supplementary Figures 4a and 4b”.

We have corrected the sentence. After the addition of new supplementary figures, the numbers have changed.

In the next sentence “We observed a similar pattern for H2B.V and H2B.” The pattern for H2B is not similar. Please adjust the text.

We apologise that the wording was not clear. We did not mean to suggest that the acetylation pattern of H2B.V and H2B are similar. Instead, we wanted to highlight that the differences between H2B and H2B.V were similar to those described for H2A and H2A.Z. We have rewritten this section to make this clear. We say now:

“Similarly, we found the N-terminal portion of H2B.V to contain a large number of highly acetylated lysines”. (page 6)

In the next paragraph, the authors say “For Histone H4 methylation, we did not observe any notable differences between TSSs and non-TSSs.” Is monomethylation of H4K2 in TSSs not a notable difference? The difference appears to be 30-40% vs 0%.

We agree with the reviewer that H4K2 mono-methylation is TSS-specific and adjusted the sentence (page 6). The sentence now reads:

“For histone H4 methylation, we did not observe notable differences between TSSs and non-TSSs, except for H4K2 monomethylation, which was TSS-specific.”

However, we did not include any quantitative statements, since the method used for quantification of methyl marks is less robust than FIPQuant.

p. 12 and Figure 6b – “No such effect was observed following HAT1 depletion.” Though clearly much less dramatic than HAT2 depletion, H2A.Z deposition or occupation does appear to be slightly reduced after HAT1 depletion. (Is figure 6b normalized to total reads?) Similarly, two paragraphs later “In contrast, depletion of HAT1 ... does not affect H2A.Z deposition” perhaps should be “In contrast, depletion of HAT1 ... only minimally affects H2A.Z deposition.”

The reviewer is correct, there is a slight effect on H2A.Z deposition following depletion of HAT1. We have made the changes suggested by the reviewer. The relevant section now reads:

“These data reveal a clear link between HAT2 depletion and H2A.Z deposition. While the time scale of the effect varied among replicates, all replicates revealed a loss of TSS-specific H2A.Z deposition (Fig. 6a and Supplementary Fig. 12). A much weaker effect was observed following HAT1 depletion (Fig. 6b and Supplementary Fig. 13).” (page 8)

“In contrast, depletion of HAT1, which results in reduced acetylation levels at H2A.Z and H2B.V, only minimally affects H2A.Z deposition.” (page 9)

Discussion – Is there any rationale for why pol II moves upstream of the TSS regions in HAT2-depleted cells? What type of nucleosomes reside there normally, or after HAT2 depletion?

p. 15, top – “

We agree that it would be interesting to know why RNA pol II moves upstream of the TSS-regions in HAT2 depletion cells. Given the apparent lack of well-defined promoter motifs, our assumption was that transcription initiation is directly affected by DNA accessibility. Since our lab has previously performed ATAC-seq experiments to evaluate DNA accessibility at other genomic loci (L. S. M. Müller et al., Nature 563, 121, 2018), we decided to use these data and to determine the degree of DNA accessibility at divergent and non-divergent TSSs.

To our surprise, we observed that the regions just upstream of the TSSs and the H2A.Z peaks were more accessible than the DNA at canonical TSSs and H2A.Z peaks. Thus, it may be that in wild type cells H2A.Z ‘directs’ the RNA pol II to the canonical sites of transcription initiation and that following depletion of HAT2 and a reduction of H2A.Z deposition, transcription initiation simply occurs at the most accessible sites, i.e. at the regions just upstream of the H2A.Z peaks and the canonical TSSs. We added a graph containing the ATAC-seq data, the H2A.Z-ChIP seq and the RNA-seq (wild type and HAT2 kd 48h) to Fig. 7 and moved the data showing the effect of HAT1 depletion to Fig. 8.

Both HAT1 and HAT2 may be evolutionary ancestors of the enzymes ... pointing to a fusion of both enzymes into one acetyltransferase during evolution.” This seems rather odd. Is there evidence that NuA4 or p400 contain two acetyltransferase domains?

It seems more likely that the substrate specificity of HAT1, HAT2, or some other HAT might change to be able to acetylate both H4 and H2A.Z, or that might be the ancestral condition, with specialization of substrates in trypanosomes. A phylogenetic tree of the of the various HATs in question seems like it might be able to confirm or dispute this model.

We feel that our statement was much too speculative and irrelevant for the other sections of the discussion. Thus, we have removed it.

Supplementary Figure 2 – H4 is nearly invisible. Is it possible to increase the contrast or intensity to make it more visible?

The gel images were taken by our collaborators simply to record the proper migration of the bands before MS analysis. Unfortunately, they are not of high quality. In some lanes (of the same sample) H4 is better visible than in others (compare lane 1 and 2). We suspect the differences to simply be related to the low image quality. In Supplementary Fig. 3, we are now showing two lanes, with lane 2 containing a band at the size of H4. Since H4 was picked up by the MS analysis, it must have been in the sample.

Supplementary Figure 13 (and others) – The colors used to depict 24 hr and 48 hr knockdowns are too similar to each other to be easily distinguished. This is important in part b, where they give very different profiles. In the legend “The data is normalized” should be “The data are normalized” (occurs twice).

We adjusted the colours used to depict 24 h and 48 h time-points in all figures and “The data is normalized” to “The data are normalized” in the legends.

Referee #2:

Main issues to be addressed:

1. The chromatin experiments that serve as the foundation for the subsequent RNAi analysis suffer from lack of detailed description, meaning the reader is left to take it on trust that everything is ok. Several aspects need to be resolved.

i). The key approach in this paper is to use TY-H2A.Z and TY-H2A immunoprecipitation to isolate nucleosomes from the TSSs and PTUs, respectively, based on the observation that the variant histone is enriched at TSSs while the core histone is depleted. The authors need to more clearly explain exactly how exactly the experiment was performed. In the results they merely state ‘By taking advantage of the

distinct genome-wide distributions of H2A.Z and H2A, we were able to enrich specifically TSS-nucleosomes (those containing H2A.Z) and non-TSS nucleosomes (those containing H2A). It is very hard work to decipher how the ‘histones were purified’ in this approach, or what level of purity was seen. Please expand on experimental descriptions in the results and methods.

The reviewer is correct, the key approach used in this paper is the genome-locus specific IP followed by mass spectrometry. We apologize for the lack of detail.

For this study, histones were isolated from TSSs, from non-TSSs and from mononucleosomes derived from whole cell lysates.

To isolate histones from TSSs we pulled on TY1-tagged H2A.Z, to isolate histones from non-TSSs, we pulled on TY1-tagged H2A. We have edited Fig. 2b to make this clearer. In addition, this approach is now clearly described in the methods section ‘locus-specific histone isolation’.

To obtain a comprehensive picture of all histone modifications, irrespectively of their genomic location, we performed an acid extraction of histones from mononucleosomes. This approach is now described in the method section: ‘Acid extraction of histones from mononucleosomes’.

To give a better overview of the different approaches, we added Supplementary Fig. 2 with an outline of the different approaches.

The purity of histones following the ‘locus-specific histone isolation’ can be seen on the gel shown in Supplementary Fig. 3. We have also added a gel to Supplementary Fig. 3 showing the histones isolated by acid extraction. The degree of histone purity was not quantified. For MS analyses, histone-containing bands were cut from a NuPAGE-Gel, adding an additional level of purification.

Equally, greater explanation is needed for what is meant by ‘fraction [%]’ and ‘acetylation degree’ in Fig2b.

We had used the terms ‘degree’ and ‘fraction’ to differentiate between values obtained from the quantitative FIPQuant and our semi-quantitative approach to estimate methylation levels (see also comments concerning subsequent questions). However, since this was obviously not very clear, we have edited the manuscript and are now using the terms acetylation [%] and methylation [%] to indicate the percentage of histones carrying an acetyl or methyl mark at a specific position. In the legends we are now stating that the acetylation percentage was determined using FIPQuant whereas the methylation percentage was estimated by spectra counting.

How was the methylation status determined (since Fig.2a only describes the assay used for identifying acetylation)? (The only explanation I could find is: ‘For a semi-quantitative analysis of histone methyl marks, we relied on peptide counts’.)

The methylation percentages were estimated by spectra counting. To this end the number of spectra for each modification type containing a specific modification site (e.g. dimethylated on K-4) were counted. The following modification forms were considered: unmodified, monomethyl, dimethyl, trimethyl. The acetylation degree of the same site was calculated using the more precise FIPQuant approach. The procedure includes $^{13}\text{C}_1$ -acetyl derivatization on the protein level to modify all unmodified lysines to $^{13}\text{C}_1$ -acetylated lysines, thereby eliminating the influence of the acetylation sites on proteolytic cleavage and on ionisation efficiencies of the generated peptides (more information can be found in our previous publication: R. ElBashir et al., Anal Chem 87, 9939, 2015).

We have added additional information on how methyl levels were estimated to the methods section: ‘Estimation of histone methylation levels’. Note, in a previous version of this manuscript, we had incorrectly written that we had used peptide counting to estimate methylation levels. We have used spectra counting to estimate methylation levels, just as we did in our previous publication (R. ElBashir et al., Anal Chem 87, 9939, 2015).

How should we consider the two forms of quantification in terms of what they can tell us, since there is little overlap in the acetylation between the two datasets in Fig2b, but considerable overlap for methylation?

Spectra counting (used to estimate the levels of histone methylation) is less precise than FIPQuant (used to determine the level of histone acetylation). Spectra counting does not consider the differences in ionisation efficiencies between differentially modified peptides (e.g. mono- and di-methylated form of the same peptide). In contrast, FIPQuant yields information on the site-specific acetylation levels from fragment ion spectra with high accuracy.

In Fig3, how were the analyses conducted, as the current description ('Lysine-specific acetylation degrees are shown for histone variants H2A.Z, H2B.V, H3.V and H4.V (blue) and were determined by FIPQuant using whole histone extracts from WT cells or from immunoprecipitation of TSS-nucleosomes') is inadequate: presumably it is impossible to use either TY-HA2 or Ty-H2A.Z to analyse H3.V and H3.V at TSSs, since they have only been reported at termination sites (see also below)? Please state clearly how each dataset was generated.

We apologise that the legend of Fig. 3 was not sufficiently clear. The PTM patterns of the canonical histones and the histone variants H3.V and H4.V were determined from whole histone extracts isolated from *T. brucei* WT cells. The PTM patterns of H2A.Z and H2B.V were determined from TY-H2A.Z immunoprecipitations. We changed the legend of Figure 3 and Supplementary Figure 7, accordingly

ii). The wording of the text suggests that TY-H2A.Z and TY-H2A immunoprecipitations are very discriminatory between TSSs and PTUs and, indeed, locus-specific (ie TSS) chromatin isolation is a key claim of the paper overall. Is this true, and how do we know this?

Yes, the ability to enrich for chromatin from specific genomic loci is an important aspect of our paper. Referee #1 has raised the same point. We consider two observations important for our 'claim' that we are able to enrich nucleosomes from distinct genomic loci.

- 1) ChIP-seq analyses of H2A.Z and H2A indicate that the distribution of these two histones is largely mutually exclusive (Fig 1a).
- 2) IPs indicate that nucleosomes containing H2A.Z have very low (undetectable) levels of H2A and that H2A-containing nucleosomes have very low (undetectable) levels of H2A.Z. We have added the co-IP data to Fig. 1b.

As outlined in our response to referee #1, even if H2A.Z and H2A distributions are not fully mutually exclusive, our ability to quantify acetylation levels should allow us to identify marks enriched at TSS or non-TSS.

Thus, taking advantage of:

- 1) the largely mutually exclusive distribution of H2A.Z and H2A,
- 2) our ability to isolate specific mononucleosomes by IP and
- 3) our ability to quantify acetyl marks

we were able to identify acetyl marks enriched at TSSs.

Identification of H4K10ac and H3K4me4 at TSSs served as a positive control. Since we had previously generated antibodies against these marks and performed ChIP-seq analyses (T. Kawahara et al., Mol. Microbiol. 69, 1054, 2008; T. N. Siegel et al., Genes & Dev. 23, 1063, 2009; J. R. Wright, et al., Mol. Biochem. Parasitol. 136, 434, 2010), we knew that these marks are enriched at TSSs. As expected from a locus-specific nucleosome enrichment, we found both marks to be enriched in TSS-nucleosomes compared to non-TSS-nucleosomes.

From Fig.2 there appears to be quite considerable data overlap in the two datasets, so how have the authors critically tested the claimed specificity?

Regarding the acetylation pattern, the overlap between TSS- and non-TSS-nucleosomes is limited to H4K4ac. Since this site is acetylated in 80% of all H4 histones (C. J. Janzen et al., FEBS Lett. 580,

2306, 2006; R. ElBashir et al., Anal Chem 87, 9939, 2015), we would not expect a specific enrichment at TSSs.

Since the patterns for all other acetyl marks are different between TSS- and non-TSS-nucleosomes, the two preparations must contain different ‘populations’ of nucleosomes. Given that they are the same two nucleosome populations that were used for the estimation of methylation levels, the considerable overlap between methylation patterns is unlikely the result of an inability to isolate different nucleosome populations.

Were the same approaches applied for all the data in Fig.3? For instance, how were H3.V and H4.V examined, since they are found at termination sites and would not be expected to be recovered in either of the TY-H2A.Z or TY-H2A immunoprecipitations?

The PTM patterns of the canonical histones and the histone variants H3.V and H4.V were determined from histones isolated by acid extraction from total mononucleosomes of WT cells. The PTM patterns of H2A.Z and H2B.V were determined for histone isolated by TY-H2A.Z immunoprecipitations. We edited the legend of Fig. 3 and Supplementary Fig. 7 to make this clear.

iii). What material was used for the acid extraction analysis shown in Fig.4?

Fig. 4 represents a summary of all modifications identified on *T. brucei* histones. Solid circles represent PTMs we have identified using acid extraction of histones from mononucleosomes OR locus-specific histone isolations. Empty circles represent PTMs we did not detect but that were previously identified by others. We have edited the legend to make this clear.

Is this the methods section entitled ‘Extraction of chromatin-associated proteins’ and, if so, in what way does this improve on the ‘technical reasons these [previous] studies failed to obtain full coverage of core histone sequences’?

Key to obtain full coverage of all histone sequences was the use of different unspecific proteases (elastase, thermolysin, papain). This is described in the results section (page 6). Previous efforts to identify PTMs in *T. brucei* relied on trypsin digestions which for many regions did not yield peptides that could be analysed by MS (C. J. Janzen et al., FEBS Lett. 580, 2306, 2006 and V. Mandava et al., Mol. Biochem. Parasitol. 156, 41, 2007).

In addition, we performed NanoLC-MS/MS analyses using an Orbitrap Fusion (Thermo Scientific) equipped with an EASY-Spray Ion Source and coupled to an EASY-nLC 1000 (Thermo Scientific), an approach that is more sensitive than the QSTAR XL QqTOF mass spectrometer (Applied Biosystems) that has been used for previous analyses.

Finally, previous studies used purified nuclei for the acid extraction. In contrast, we used mononucleosomes, a strategy which may have further improved the purity.

The methods section outlining the ‘Extraction of chromatin-associated proteins’ describes the protocol we used to differentiate chromatin bound histones (insoluble histones) from non-deposited histones (free, soluble histones). The material generated using this approach was only used for Fig. 6c and Fig. 8b.

2. HAT2. The combined data in Figs 5, 6 and 7 is interpreted to suggest that RNAi leads to a specific loss of histone H4 acetylation on lysines 2, 5 (perhaps) and 10, leading to altered deposition of histone H2A.Z and RNA Pol II, with a small change in transcription initiation site but no clear loss of mRNA. This is a nice set of data, but is it complete? One simpler explanation, not explored and less mechanistically clean, is that the RNAi has a dramatic effect on overall levels of H2A.Z, which could account for the ChIPseq and might explain the lack of specific Pol II localisation at the TSSs. Have they run western on whole cell extracts after RNAi?

Yes, we performed western blot analyses on whole cell lysates following HAT2 depletion for 48 h. The blots indicate that the overall H2A.Z levels are not affected by the depletion of HAT2 (see Figure 1, below).

Since our H2A.Z ChIP-seq analyses only provide information on the relative position of the histone variant along the genome, not on the total amount of H2A.Z deposited, lower H2A.Z levels should not result in a decrease of H2A.Z peaks as we have observed them by ChIP-seq. We have tried to illustrate our reasoning in the lower panel of Fig. 1.

Thus, combined our data suggest that HAT2 depletion directly affects H2A.Z deposition and that the reduced H2A.Z levels at TSS are not simply the consequence of lower total H2A.Z levels.

Figure 1: a) western blot showing H2A.Z levels following HAT2 depletion for 48 h. b) illustration of how lower H2A.Z would affect our ChIP-seq results.

Secondly, do we know that HAT2 RNAi has the stated, highly specific effect on histone H4, and that this directly leads only to the effects described on H2A.Z and Pol II? Did the authors run acetylation assays in the RNAi cells relative to WT to look at all histones?

Yes, the four canonical histones and their corresponding histone variants were included in all acetylation assays performed for this study. We could not detect an effect of HAT1 or HAT2 depletion on any histone except for H2A.Z, H2B.V and H4. We included Supplementary Fig. 11 to show acetylation levels of H2A, H2B, H3, H3.V and H4.V following HAT1 or HAT2 depletion for 48 h.

3. HAT1. Here, the authors show that RNAi leads to widespread effects on acetylation of H4, H2A.Z and H2B.V, with associated loss of mRNA that cannot be explained by altered H2A.Z localisation. While it is true that this suggests a distinct response to HAT1 RNAi, the experiments do not provide a mechanistic explanation and, at least superficially, the findings seem comparable to data on Esa1 and Tip60 function in yeast and humans. Greater discussion of this overlap is warranted.

As the reviewer correctly states, our data suggest distinct functions of HAT1 and HAT2. The two enzymes appear to acetylate different residues and depletion of HAT2 has a much stronger effect on deposition of H2A.Z deposition than depletion of HAT1. In addition, while depletion of HAT1 affects total mRNA levels, depletion of HAT2 appears to affect the site of transcription initiation but not total

mRNA levels. We are not sure whether the reviewer is asking for a greater discussion of the overlap between HAT1 and HAT2 or for a discussion of the overlap between HAT1 and Esa1 / Tip60. To address the reviewer's concerns, we have tried to address both points in more detail.

We agree that there is some overlap between the roles of HAT1 and Esa1/Tip60. Just like HAT1, Esa1/Tip60 acetylate htz/H2A.Z and H4. However, while Esa1 appears to be responsible for the bulk of H4 acetylation our data suggest that in *T. brucei* loss of HAT2 has a stronger effect on H4 acetylation than HAT1. In addition, while Esa1-mediated acetylation of histones was shown to increase deposition of htz in yeast, our data suggest that depletion of HAT1 had a very minor effect on H2A.Z deposition in *T. brucei*. Thus, there are clear differences between the roles of HAT1 and Esa1/Tip60. The differences and similarities are now outlined in the discussion.

Regarding the mechanism by which HAT1 may affect transcription, our RNA pol II chip data showed no large differences between wild type cells and HAT1 depleted cells. Yet, overall mRNA levels are 50% lower. To us these observations suggest that loss of H2A.Zac, induced by depletion of HAT1, affects the recruitment of RNA pol II rather than the rate of transcription. (An effect on the rate of transcription would have changed the pattern along the genome.) However, the overall amount of DNA (or chromatin) associated RNA pol II cannot be inferred from ChIP-seq (unless spike-ins are used). Thus, to determine if depletion of HAT1 affects RNA pol II recruitment, we isolated chromatin and determined the amount chromatin-associated RNA pol II by western blotting (Fig. 8b). The new data suggest that following HAT1 much less RNA pol II associated with chromatin than in wild type cells. Thus, it is possible that H2A.Zac is important for efficient RNA pol II recruitment to TSSs. We have added the new data to the results section and extended the discussion to mention this possibility.

Moreover, why do they authors discount the possibility that HAT1 and HAT2 arose by gene duplication in trypanosomes to provide specialised functions? At the moment they seem to think all (or most) other eukaryotes have undergone an HAT1-2 gene fusion, which seems a more far-fetched explanation:

As stated in the response to referee #1, we agree with the referee's concerns and feel that our statement was much too speculative and irrelevant for the other sections of the discussion. We have thus removed it.

Referee #3:

This reviewer would like to see the statistical analysis associated with the measurements in Figure 5 (with p-values associated with the differences) - this is a minor revision.

As recommended by the reviewer, we have added statistical analyses to all figures showing the impact of the two HATs on acetylation levels. Details on the statistical analysis have been added to the methods in the Supplementary Information: "Statistical analysis of the impact of HAT depletion on histone acetylation levels". The relevant section reads as follows:

Statistical analyses of the impact of HAT depletion on histone acetylation levels were performed using GraphPad Prism version 7.0c. We applied a multiple t-test between the different conditions, with three replicates per condition. Individual p-values for each lysine position were computed using the two-way ANOVA approach recommended for comparing samples from different conditions https://www.graphpad.com/guides/prism/7/statistics/index.htm?stat_options_for_multiple_t_tests.htm. The statistical significance was determined using the Holm-Sidak method to correct for multiple comparison. Adjusted p-values < 0.05 were defined as 'statistically significant' and marked with asterisk in the relevant figures (Supplementary Table 7).

Reviewers' Comments:

Reviewer #2:

Remarks to the Author:

The authors are to be commended for the rigour of their extensive responses to my, and other, comments. In my opinion the manuscript has been substantially improved.

Response to Reviews

Referee #2:

The authors are to be commended for the rigour of their extensive responses to my, and other, comments. In my opinion the manuscript has been substantially improved.

We thank the reviewer for the thoughtful feedback and are happy to have addressed all remaining comments.